# Monitoring CO emissions of the metropolis Mexico City using TROPOMI CO observations

Tobias Borsdorff[1], Agustín García Reynoso[2], Gilberto Maldonado[2], Bertha Mar-Morales[2],
Wolfgang Stremme[2], Michel Grutter[2], and Jochen Landgraf[1]

[1]Netherlands Institute for Space Research, SRON, Utrecht, the Netherlands
[2]Centro de Ciencias de la Atmósfera, Universidad Nacional Autónoma de México, México City, Mexíco

**Correspondence:** T. Borsdorff (t.borsdorff@sron.nl)

**Abstract.** The Tropospheric Monitoring Instrument (TROPOMI) on ESA Copernicus Sentinel-5 satellite (S5-P) measures carbon monoxide (CO) total column concentrations as one of its primary targets. In this study, we analyze TROPOMI observations over Mexico City in the period 14 November 2017 to 25 August 2019 by means of collocated CO simulations using the regional Weather Research and Forecasting (WRF-chem) model. We conclude on the emissions from different urban districts in the region. Our WRF-chem simulation distinguishes CO emissions from the districts Tula, Pachuca, Tulancingo, Toluca, Cuernavaca, Cuautla, Tlaxcala, Puebla, Ciudad de Mexico, and Arena Ciudad de Mexico by 10 separate tracers. For the data interpretation, we apply a source inversion approach determining per district the mean emission and the temporal variability, latter regularized to reduce the propagation of the instrument noise and forward model errors in the inversion. In this way, the TROPOMI observations are used to evaluate the "Inventario Nacional de Emisiones de Contaminantes Criterio" (INEM) inventory that was adapted to the period 2017-2019 using in-situ ground-based observations. For the Tula and Pachuca urban areas in the North of Mexico City, we obtain $0.10 \pm 0.004$ Tg/yr and $0.09 \pm 0.005$ Tg/yr CO emissions, which exceeds significantly the INEM emissions <0.008 Tg/yr for both areas. On the other hand for Ciudad de Mexico, TROPOMI estimates emissions of $0.14 \pm 0.006$ Tg/yr CO, which is about half of the INEM emissions of 0.25 Tg/yr and for the adjacent district Arena Ciudad de Mexico the emission is $0.28 \pm 0.01$ Tg/yr according to TROPOMI observations versus 0.14 Tg/yr as stated by the INEM inventory. Interestingly, the total emission of both districts is similar ($0.42 \pm 0.016$ Tg/yr TROPOMI versus 0.39 Tg/yr adapted INEM emissions). Moreover, for both areas we found that the TROPOMI emission estimates follow a clear weakly cycle with a minimum during the weekend. This agrees well with ground-based in-situ measurements from the "Secretaria del Medio Ambiente" (SEDEMA) and Fourier Transform Spectrometer column measurements in Mexico City that is operated by the Network for the detection of Atmospheric Composition Change Infrared Working Group (NDACC-IRWG). Overall, our study demonstrates an approach to deploy the large amount of TROPOMI CO data to conclude on urban emissions on sub-city scales for metropolises like Mexico City. Moreover, for the exploitation of TROPOMI CO observations our analysis indicates the clear need for further improvements of regional models like WRF-chem, in particular with respect to the prediction of the local wind fields.

# 1 Introduction

Carbon monoxide (CO) is an atmospheric trace gas emitted by incomplete combustion to the atmosphere (e.g. biomass burning, industrial activity, and traffic). Its background concentration is relatively low with an atmospheric residence time varying from days to month (Holloway et al., 2000) depending on the atmospheric concentration of the hydroxyl radical (Spivakovsky et al., 2000). These characteristics established CO as a tracer for air pollution and transport processes in the atmosphere (e.g. (Gloudemans et al., 2009; Pommier et al., 2013; Schneising et al., 2019)).

The Tropospheric Monitoring Instrument (TROPOMI) launched 2017 as single payload of ESA's Copernicus Sentinel-5 Precursor mission aims on CO as one of its primary targets. The operational CO column product is inferred from TROPOMI's shortwave infrared measurements with daily global coverage and a high spatial resolution of 7x7 $km^2$ (Veefkind et al., 2012). Early in the mission, the TROPOMI CO dataset was validated with ground-based measurements of the Total Carbon Column Observing Network (TCCON) (Borsdorff et al., 2018a), and inter-compared with simulated CO fields of the European Centre for Medium-Range Weather Forecasts (ECMWF) - Integrated Forecasting System (Borsdorff et al., 2018b). On 11 July 2018, it was concluded that the TROPOMI CO data quality is fully compliant with the mission requirements of 15% precision and 10% accuracy and so it was released for public usage (https://scihub.copernicus.eu).

Borsdorff et al. (2018a, 2019) illustrated the capability of TROPOMI to detect CO emissions from pollution hot spots of medium size to large cites (e.g. Yerevan, Tabriz, Urmia, and Tehran), industrial areas (e.g. Po valley in Italy), and even pollution along arterial roads in Armenia. To monitor the emissions of metropolises, data interpretation of multi-annual data sets is required. The different inversion techniques discussed by (Varon et al., 2018) for plume inversions, i.e. the source pixel method, the mass balance method and the inversion of a Gaussian plume model are appropriate to interpret emission of point sources but are less suitable for flux inversion of spatially extended sources. Therefore, in this study we estimate CO emission by inverting simulations of the regional atmospheric modeling Weather Research and Forecasting (WRF-chem) as an atmospheric tracer transport model, which allows to simulate the CO column on the spatial resolution as TROPOMI. Possible error sources of this type of flux inversion is the limited validity of the simulated wind fields, prior assumption on the spatial distribution of emissions, and the simulated atmospheric dispersion (Borsdorff et al., 2019).

Mexico City is a prime example of a CO pollution hot spot that is clearly detectable by TROPOMI. It is a fast-growing mega city located at an altitude of 2240 m on the Central Plateau which is surrounded by mountains. The urban area is divided in ten different urban districts (Tula, Pachuca, Tulancingo, Toluca, CdMx, Cuernavaca, Cuautla, Tlaxcala, Puebla, Ciudad de Mexico (CdMx), and Arena Ciudad de Mexico (ACdMx)) and the metropolis has a long history of atmospheric pollution measurements. More than 29 in-situ CO measurements stations are distributed over the city operated by the "Secretaria del Medio Ambiente" (SEDEMA, Mexican Ministry of the Environment). About every two years, the ministry reports on the CO emission of Mexico City. Based on the bottom-up approach using the in-situ measurements, it is concluded that a major part of Mexico City's CO emission is caused by light duty motor vehicles with a significant decline in the recent years. For the Zona metropolitana del valle de Mexico (ZMVM), SEDEMA reported a reduction of CO emissions from 2.04 to 0.7 and 0.28 Tg/yr in the years 2000, 2014, and 2016 SMA-GDF (2018).

These in-situ measurements are complemented by ground-based FTIR observations of the NDACC (Network for the detection of Atmospheric Composition Change) - IRWG (Infrared Working Group) network, which among other products provide regularly CO total column concentrations. Using NDACC and IASI satellite observations of CO, Stremme et al. (2013) estimated the overall annual CO emission of Mexico City to be about 2.15 Tg/yr for the year 2008. Building on this, TROPOMI CO observations add new possibilities for air quality monitoring due to the regional coverage, the daily overpass combined with the high precision of the data.

In this study, we analyze about two years of TROPOMI CO measurements using collocated WRF-chem CO simulations for Mexico to get more insight into the emission of Mexico City. To this end, in Section 1 we introduce the TROPOMI CO dataset and the simulation of the WRF-chem model and Section 2 describes our methodology to fit the WRF-chem model to the TROPOMI data for emission estimates. Sections 3 discusses our finding and section 4 gives the summary and conclusion.

## 2   TROPOMI CO data set

To investigate CO emission of the Mexico City metropolis, we select the TROPOMI dataset of CO total column observations between 14 November 2017 and 25 August 2019 over Mexico. The data are processed with the shortwave infrared CO retrieval (SICOR) algorithm that was developed for the Copernicus operational data processing (Landgraf et al., 2016a). Algorithm settings like the spectral windows, priori profiles and other auxiliary data are reported by Landgraf et al. (2016b). The SICOR algorithm accounts for for atmospheric scattering by retrieving effective cloud parameters (altitude, optical thickness) together with the total column concentrations of CO and of the interfering gases $H_2O$, HDO and $CH_4$. The radiative transfer simulation uses the HITRAN 2016 database for all species as described by Borsdorff et al. (2019) and the inversion deploys the profile scaling approach that scales a reference profile to fit the spectral measurement (Borsdorff et al., 2014). Here, the priori profile is taken from a spatio-temporally resolved atmospheric transport simulations of the TM5 model (Krol et al., 2005). The TROPOMI CO data product includes the total column averaging kernel $a_{col}$ that relates the true vertical CO profile $\rho_{true}$ to the retrieved total column concentration $c_{ret}$ following the equation

$$c_{ret} = a_{col}\rho_{true} + \epsilon \tag{1}$$

with the noise contribution $\epsilon$. This study limits the analysis to scenes under clear-sky and low-cloud atmospheric conditions. This corresponds to quality assurance value $q > 0.5$ which is also provided by the S5P data product. Finally, the individual TROPOMI CO orbits show an artificial striping in flight direction, probably due to a deficient instrument calibration. To reduce this feature, we apply an a posteriori data correction as discussed by (Borsdorff et al., 2019) based on frequency filtering in the Fourier space. Finally, on 5 August, 2019, the spatial sampling of the data product at satellite nadir geometry was improved from 7x7 $km^2$ to 7x5.6 $km^2$ due to a shorter readout time of the detectors. This event is covered by our data set.

## 3 Methodology

### 3.1 The WRF-chem model

We simulate the TROPOMI CO column concentrations by deploying the WRF-chem model version 3.9.1.1. The simulation covers the time period of TROPOMI measurements on the regional domain shown in Fig. 1. We ignore photo-chemical oxida-
tion and secondary production of CO in the atmosphere (chem_opt option 106 (RADM2-KPP), as a tracer with gaschem off), which is justified by the long lifetime of CO compared with the size of the model domain as discussed by Dekker et al. (2017). Especially, for the region of Mexico City the contribution of atmospheric chemistry to the total CO concentration is less than 3% as presented by Mejia (2020). Hence, WRF-chem simulates the transport of CO surface emission as traces as done by e.g. Borsdorff et al. (2019); Dekker et al. (2017, 2018). The spatial resolution of the simulation is chosen to be comparable with the
TROPOMI CO product sampling. Each grid cell of the considered simulation domain (270x270 km$^2$) is 3x3 km$^2$. The WRF-chem simulation employs the emission inventory "Inventario Nacional de Emisiones de Contaminantes Criterio" (INEM) for the year 2013 but scaled by a factor of 0.48 to make it applicable to the years 2017 to 2019. Here the scaling factor is based on recent SEDEMA surface measurements (García-Reynoso et al., 2018). The inventory includes diurnal, week-to-week and monthly variations of the CO emissions, where weekly and daily temporal profiles are derived from traffic counts in Mexico.
The inventory is described in more detail by (García-Reynoso et al., 2018). Finally, the model run is constraint by NCEP North American Mesoscale (NAM) 12 km analysis wind fields (NCEP, 2015) and yields vertical CO concentration profiles for every latitude/longitude grid cell and every model time step and tracer run.

The WRF-chem simulation uses ten independent tracers to estimate the CO emissions of the areas Tula, Pachuca, Tulancingo, Toluca, Cuernavaca, Cuautla, Tlaxcala, Puebla, the metropolian area of Mexico City CdMx, and the adjoint urban area ACdMx.
The total simulated CO field is given by the sum of the simulated CO fields of the tracer. Since no atmospheric chemistry is accounted, each CO tracer field is linear in a the corresponding emissions per district,

$$F_{\text{WRF}}(\alpha_1, \cdots, \alpha_{10}) = \sum_{i=1}^{10} \mathbf{k}_i \alpha_i \tag{2}$$

where $\alpha_i$ is the corresponding scaling factor and $\mathbf{k}_i$ represents the CO tracer field for the reference emission (adapted INEM data) for $\alpha_i = 1$.
Before using our model to simulate TROPOMI data, we interpolate the model fields to the geolocation and time of the TROPOMI observations. Subsequently, we integrate the model CO profiles to total column densities by applying the total column averaging kernel of the TROPOMI CO retrieval following equation 1. We summarize this numerical step in the observation operator $\mathcal{O}$, which transforms the forward model into

$$F_{\text{sat}}(\alpha_1, \cdots, \alpha_{10}) = \sum_{i=1}^{10} \mathcal{O}(\mathbf{k}_i) \alpha_i \tag{3}$$

Hence, the operator $\mathcal{O}$ accounts for the TROPOMI specific vertical sensitivity, which can change from measurement to measurement and so ensures that the comparison between TROPOMI and WRF-chem is free of the null-space or smoothing error

(Rodgers, 2000; Borsdorff et al., 2014). Here, the scaling factors $\alpha_i$ are not affected by the operation. In a next step, we transform Eq. (3) to

$$F_{\text{sat}}(E_1, \cdots, E_{10}) = \sum_{i=1}^{10} \mathcal{O}(\tilde{\mathbf{k}}_i) E_i \tag{4}$$

where $\tilde{\mathbf{k}}_i = \frac{\mathbf{k}_i}{E_{i,INEM}}$ and $E_i = \alpha_i E_{i,INEM}$ with the corresponding emissions $E_{i,INEM}$ of the INEM inventory interpolated to the TROPOMI overpass time.

To improve the capability of our forward model to fit TROPOMI observations, we introduce a spatially constant CO background field $\mathbf{k}_{\text{bg}}$ and an altitude dependence term $\mathbf{k}_{\text{elv}} = z - z_{\text{ref}}$ with corresponding scaling factors $\alpha_{\text{bg}}$ and $\alpha_{\text{elv}}$. Here, $z$ is the respective elevation of the TROPOMI CO ground pixels and $z_{\text{ref}} = 2240$ m is an arbitrary reference altitude set to the elevation of Mexico City,

$$F_{\text{sat}}(E_1, \cdots, E_{10}, \alpha_{\text{bg}}, \alpha_{\text{elv}}) = \sum_{i=1}^{10} \mathcal{O}(\tilde{\mathbf{k}}_i) E_i + \mathbf{k}_{\text{bg}} \alpha_{\text{bg}} + \mathbf{k}_{\text{elv}} \alpha_{\text{elv}} . \tag{5}$$

These two effective model components account for CO contribution over the Mexico City area originating from outside the model domain such as fires, power plants, biogenic production, other cities as well as the long range transport (Borsdorff et al., 2019) and an altitude dependent linear vertical gradient of the CO columns. Both do not interfere with any localized emission sources. They mitigate shortcomings of the WRF-chem simulations ignoring CO boundary conditions at the model domain.

Finally, for the interpretation of our CO forward simulations, we make an important assumption. Although the WRF-chem simulations account for the temporal accumulation of the localized CO emission over days and weeks, we allocate an emission estimate of the corresponding overpass time to each TROPOMI overpass. Here, we assume that a TROPOMI CO image is dominated by the emissions of the urban districts for the particular observation day, where the temporal accumulation of CO from previous days is partly described by the WRF-chem simulation due to the corresponding scaling of the inventory and partly mitigated by fitting the nuisance parameter $\alpha_{\text{bg}}$ and $\alpha_{\text{elv}}$.

### 3.2 Inversion methodology

Interpreting a series of $n$ TROPOMI CO images

$$y = (y_1, \cdots, y_n) \tag{6}$$

at overpass times $t_0, \cdots, t_n$ means to estimate the corresponding emissions given by the state vector

$$x = (x_1, \cdots, x_n) , \tag{7}$$

where each element comprises

$$x_i = (E_{1,i}, \cdots, E_{10,i}, \alpha_{\text{bg},i}, \alpha_{\text{elv},i}) \tag{8}$$

at the corresponding time $t_i$. Our linear forward model in Eq. (5) describes the measurement vector y by

$$\begin{pmatrix} y_1 \\ y_2 \\ \vdots \\ y_n \end{pmatrix} = \begin{pmatrix} \mathbf{K}_1 & 0 & \cdots & 0 \\ 0 & \mathbf{K}_2 & \cdots & 0 \\ \vdots & \vdots & \ddots & \vdots \\ 0 & 0 & \cdots & \mathbf{K}_n \end{pmatrix} \begin{pmatrix} x_1 \\ x_2 \\ \vdots \\ x_n \end{pmatrix} \tag{9}$$

with the forward model Jacobian $\mathbf{K}_i = (\mathcal{O}(\tilde{\mathbf{k}}_{1,i}), \cdots \mathcal{O}(\tilde{\mathbf{k}}_{10,i}), \mathbf{k}_{\mathrm{bg},i}, \mathbf{k}_{\mathrm{elv},i})$, in short $y = \mathbf{K}x$. Equation (9) can be inverted by

$$x_{\mathrm{est}} = \min_x \left\{ ||y - \mathbf{K}x||^2_{\mathbf{S}_e} \right\} , \tag{10}$$

5    which is equivalent to the solution $(x_{\mathrm{est},1}, \cdots, x_{\mathrm{est},n})$ of the individual problems $y_i = \mathbf{K}_i x_i$ due to the block diagonal form of Eq. (9). Here, the norm of an arbitrary vector $p$ is defined by $||p||^2_{\mathbf{S}_e} = p^T \mathbf{S}_e^{-1} p$ and $\mathbf{S}_e$ is the measurement error covariance matrix with the variance of the TROPOMI retrieval error on the diagonal.

     Due to measurement noise and forward model errors, the least squares inversion of Eq. (10) results in unfavorable error propagation and so requires regularization. Because our problem is linear in the state vector $x$ regularization can be performed 10    as part of the fitting approach or a posteriori to the least squares solution, without loss of generality. To regularize the noise propagation, we first derive the temporal mean

$$\bar{x}_{\mathrm{est}} = \frac{1}{n} \sum_{i=1}^{n} x_{\mathrm{est},i} \tag{11}$$

from the non-regularized solution in Eq. (10). This modifies our cost function to

$$x_{\mathrm{est}} = \min_x \left\{ ||y - \mathbf{K}x||^2_{\mathbf{S}_e} \right\} \quad \text{with} \quad \bar{x} = \frac{1}{n} \sum_{i=1}^{n} x_i \tag{12}$$

15    In this way, we divided the solution of the original inversion problem (10) in two steps: First we determine the mean emission from the individual least squares solutions $x_{\mathrm{est},i}$, which yields the constrained least squares problem in Eq. (12) to describe the temporal variability. The side constraint guarantees that measurement information is not used twice. Finally, we add an additional Tikhonov side constraint to Eq. 12 to regularize the error propagation,

$$x_{\mathrm{est}} = \min_x \left\{ ||y - \mathbf{K}x||^2_{\mathbf{S}_e} + ||x - \bar{x}||^2_{\boldsymbol{\Gamma}} \right\} \tag{13}$$

20    with

$$\bar{x} = \frac{1}{n} \sum_{i=1}^{n} x_i . \tag{14}$$

Here, $\boldsymbol{\Gamma}$ is an appropriate regularization matrix. For a block diagonal form of $\boldsymbol{\Gamma}$ analogous to the Jacobian in Eq. (9), namely

$$\boldsymbol{\Gamma} = \begin{pmatrix} \boldsymbol{\Gamma}_1 & 0 & \cdots & 0 \\ 0 & \boldsymbol{\Gamma}_2 & \cdots & 0 \\ \vdots & \vdots & \ddots & \vdots \\ 0 & 0 & \cdots & \boldsymbol{\Gamma}_n \end{pmatrix} \tag{15}$$

the minimization problem (13) decomposes into $n$ problems

$$x_{\text{est,i}} = \min_{x_i} \left\{ ||y_i - \mathbf{K_i}x_i||_{\mathbf{S_e}}^2 + ||x_i - \bar{x}||_{\mathbf{\Gamma}}^2 \right\} \tag{16}$$

which are only coupled by the external side constraint (14). A closer look at our inversion problem shows that the two constraints have similar effects. The Tikhonov constraint $||x - \bar{x}||_{\mathbf{S_e}}$ minimizes the variation of the state vector around its mean depending on the regularization parameter $\lambda$, whereas the external constraint requires strict conservation of the mean.

Therefore, in practice, we solve the inversion (16) and evaluate the external constraint on the mean afterwards to confirm proper use of the measurement information. Its solution is given by

$$x_{\text{est,i}} = \mathbf{G}_i(y_i - \mathbf{K}_i\bar{x}_i) + \bar{x}_i \tag{17}$$

with the gain matrix

$$\mathbf{G}_i = (\mathbf{K}_i^T \mathbf{S}_{\text{e,i}}^{-1} \mathbf{K}_i + \mathbf{\Gamma}_i)^{-1} \mathbf{K}_i^T \mathbf{S}_{\text{e,i}}^{-1} \tag{18}$$

The inversion's averaging kernel relates the 'true' state vector $x_{\text{true,i}}$ to $x_{\text{est,i}}$, namely

$$x_{\text{est,i}} = \mathbf{A}_i (x_{\text{true,i}} - x_i) + \bar{x} \tag{19}$$

with

$$\mathbf{A}_i = \mathbf{G}_i \mathbf{K}_i \tag{20}$$

$\mathbf{A}_i$ represents the derivative $\mathbf{A}_{\text{i,kl}} = \frac{\partial x_{\text{est},j}}{\partial x_{\text{true},l}}$, where its diagonal elements describe the retrieval sensitivity of a state vector element to its true value. The degree of freedom for signal

$$\text{DFS}_i = \text{trace}(\mathbf{A}_i), \tag{21}$$

indicates the total number of independent pieces of information.

To evaluate the fit quality for each overpass, we consider the fit residuals $\delta_i = y_i - \mathbf{K}_i x_{\text{est},i}$. Additionally, we evaluate the goodness of the fit described by the reduced chi squared value,

$$\chi_i^2 = \frac{1}{\nu_i} \sum_{l=1}^{L} (\delta_i, l/y_{\text{err},ik})^2 . \tag{22}$$

Here $L$ is the number of observation of a single overpass, $y_{\text{err},i}$ the retrieval error, and $\nu_i = L - \text{DFS}_i$.

### 3.3 Estimate of the mean emissions

The first step of the inversion described in the previous section means to determine the prior emissions from a set of TROPOMI data with highest information content using a non-regularized least squares fit, $\mathbf{\Gamma} = 0$. Here, the individual emission estimates may be noisy due to enhanced error propagation in the inversion, however, averaging all inversions reduces noise contribution

and so gives a reliable estimate of a mean emission for the different districts. The validity of this approach highly depends on the selected data set of TROPOMI overpasses. On the one hand, the ensemble should be large enough to estimate mean emissions for the considered time period, but on the other hand it should be strictly filtered for cases where the forward model is in good agreement with the measurement such that a stable inversion of the all emissions is possible. The information content of a single overpass varies and depends on several aspects: (1) The number of useful measurements and their cloud coverage changes between different TROPOMI overpasses. Here, clouds shield the lower troposphere, where atmospheric measurements are particular sensitive to the surface emissions $E_i$. (2) The pixel size at the swath edge is about 32 km and so about 5 times larger than at the sub-satellite point. This reduces not only the number of pixels covering a certain area but also the sensitivity of the individual TROPOMI observations. (3) The quality of the forward model depends on the meteorological situation, where we consider model simulations for low wind speeds more reliable. These considerations led to the criteria of the data filtering to determine the mean emission for each district. We only select overpasses which meet both filter criteria:

- 70 % of the data domain is covered by TROPOMI observations

- for all observations the across track pixel size is $< 15$ km.

The filter criteria reduce the original set of 551 overpasses to 199, which we consider to be sufficient to estimate the overall average emission rate per district, yielding $\bar{x}$. For this we use the median instead of the mean because of its robustness against outliers. With the same reasoning we define the percentile difference

$$\delta P_j = |\frac{P_j(84.1) - P_j(15.9)}{2}| \tag{23}$$

, to describe scattering in the data, which corresponds to the standard deviation of normal distributed parameters. Finally, we calculate the error of the mean using the percentile difference.

## 3.4 Final data product

Subsequently, the final data reduction step is performed solving the inversion problem (13). For all overpasses, we choose $\mathbf{\Gamma}_i$ to be a diagonal matrix with

$$\text{diag}\,\mathbf{\Gamma}_i = [\gamma_1, \gamma_2, \cdots, \gamma_{10}, 0, 0] \tag{24}$$

where the zeros ensure that the elements of the state vector $\alpha_{\text{bg}}$ and $\alpha_{\text{elv}}$ are not regularized. Obviously, the regularization parameter $\gamma_k$ must be well-chosen to optimize the balance between minimum error propagation on the fit parameter and maximum information content inferred from the measurement. If $\gamma_k$ is chosen too small, the propagation of the TROPOMI measurement noise as well as retrieval biases and forward model errors dominates the inversion. If $\gamma_k$ is chosen too large, the estimated state vector reproduces the prior estimate without appropriate use the information content of the measurement. For our application, we fix the regularization parameter $\gamma_k$ for $k = 1, \cdots, 10$ to constant values such that the scatter of the retrieved emissions stays within predefined boundaries.

Considering the temporal variation of the INEM emissions to be about $40\%$, we adjusted the regularization parameter $\gamma_1, \cdots, \gamma_{10}$ such that the retrieved emissions vary within $60\%$ around their average. The value 60% is empirical chosen to balance information content against noise propagation. It puts a moderate constraints on the inversion ensuring on the one hand a stable inversion and on the other hand a realistic variation of the retrieved emissions around the priori.

One great advantage of the final retrieval product is that it includes the averaging kernel $\mathbf{A}_i$. This can be used to filter the data with respect to the information provided by the TROPOMI measurements. For each tracer emission, we filter on the individual emission $E_l$, considering averaging kernel values $\mathbf{A}_{l,l} > 0.3$. This form of data mining optimizes the data use, keeping in mind that TROPOMI overpasses may be appropriate to determine one specific source but not all sources simultaneously. The concept of information content based filtering turned out to be very useful and enhances the data exploitation compared to the non-regularized least squares fitting used to determine the mean emission values.

## 4   Results

Fig. 2 shows the CO background that was fitted as an auxiliary parameter during the inversion described in Sec. 3.2. The concentration and its annual cycle are shown. Here, the biomass burning season between February and June causes the corresponding CO enhancement, whereas lower CO concentrations are observed during the rainy season between June and November. The extremely high CO column values on the 15th May 2019 are due to the transport of CO enriched air from wild fires in the South-West of Mexico in to the model domain. Figure 3 shows the CO concentration in the state of Mexico under normal conditions and after the fires, which caused a serious health hazard in Mexico City. These type of fires outside the model domain create an inhomogeneous background CO field over Mexico City, which cannot be described by our forward model. Only fitting a constant background is not sufficient in these extreme cases and so during the fire season many data cannot be used (we excluded the month May and June 2019).

Figure 4 shows three examples of TROPOMI overpasses, which includes a pixel resolution of 7x7 km$^2$ (panel (a), (b), and (c)) and the enhanced spatial resolution of 7x5.6 km$^2$ (panel (d)), where latter is the TROPOMI instrument baseline since the 6th of August 2019. Focusing on the dry season, the TROPOMI instrument can detect distinct CO enhancements over the different emission areas in Central Mexico with the retrievals from single orbit overpasses (see left column of Fig. 4). After fitting our forward model to the TROPOMI measurements brings simulated data and observations into good agreement as illustrated in Fig. 4. Particular for low wind speed conditions in Fig. 4(a), TROPOMI and WRF-chem show distinct CO enhancements over the different emission areas of Mexico. Furthermore, the transport of CO enhanced air form Mexico City towards the South following the mountain orography and the accumulation of CO in the South is seen by TROPOMI in agreement with the WRF-chem simulation (4(c). This clearly shows that regional models like WRF-chem have a great potential to reproduce the large-scale patterns seen by the TROPOMI instrument. However, we also found clear localized residuals in the difference $\delta_j$ between observations and forward model. (right column of Fig. 4). For atmospheric conditions under high wind speeds the WRF-chem simulations can deviate more from the TROPOMI measurements as shown in Fig. 4 (c). Here, the plume of CO enriched air extending from Mexico City towards the North is simulated very narrow compared to the more dispersed plume

seen by TROPOMI. This points to a possible underestimation of the atmospheric dispersion in the WRF-chem simulation. A very prominent residual between TROPOMI and WRF-chem is shown in 4 (d) but also present in 4 (a) and (b). Here TROPOMI measures a strong CO enhancement in the North of Mexico City that is not reproduced by the WRF-chem model. This points at a deficient spatial distribution of INEM emissions.

For each tracer domain Fig. 5 (a) shows the mean emissions of the the adapted INEM inventory, the non-regularized least squares fit and the final data product. The mean emissions agree very well between the last two approaches, indicating that the final inversion in Eq. 17 satisfies the constraint of the pre-defined mean value. This supports our assumption that the external constraint does not have to be accounted for the chosen Tikhonov constraint of the inversion. The scatter of the least squares product is high and in most cases exceeds 100% (see Fig. 5 (b)), which is expected for the non-constraint inversion.

Moreover, we find significant differences between the emissions of the prior and the final data product. The retrieved emissions for the urban districts Tula ($0.10 \pm 0.004$ Tg/yr) and Pachuca ($0.09 \pm 0.005$ Tg/yr) in the North of Mexico City seem to be underestimated by the emission inventory (both were less then 0.008 Tg/yr). It is not yet clear what sources are missing in the inventory, this needs to be addressed in future studies. However, we identified an oil refinery and a power plant near to Tula and cement and lime kilns near to Pachuca that could contribute to the CO emissions. Furthermore, we found that the emission

of the central part of Mexico City (CdMx) is assumed too high in the adapted INEM inventory (0.25 Tg/yr), where TROPOMI measurements indicate lower values for CdMx ($0.14 \pm 0.006$ Tg/yr). This comes along with higher values for the adjacent district ACdMx ($0.28 \pm 0.01$ Tg/yr). The sum of both emissions ($0.42 \pm 0.016$ Tg/yr) is similar to the priori emissions (0.39 Tg/yr). This may mean that the total emissions of the domain including CdMx and ACdMx is well represented in the emission inventory but only the spatial distribution of the source intensity needs refinement.

The $\chi^2$ values in Fig. 5 clearly show that the agreement between TROPOMI and WRF-chem can be improved by fitting the emissions of the different city districts (blue line) instead of using the INEM inventory (grey line). The regularization approach increases the $\chi^2$ values (green bars) because the inversion can less compensate differences between TROPOMI and WRF-chem by choosing unrealistic emissions. However, the $\chi^2$ values of the Final-Fit are still lower than the ones for the prior INEM emissions (grey line). Overall, the $\chi^2$ values exceeds 1 which indicates that the difference between TROPOMI and WRF-chem

is dominated by systematic errors in the WRF-chem simulation. Figure 6 shows the $\chi^2$ values for the difference between the TROPOMI CO observations of single overpasses and the WRF-chem simulations over the considered time range of the study. The $\chi^2$ values follow a seasonal pattern with enhanced values during the biomass burning season between February and June and low values during the rainy season between June and November. As mentioned before, in the vicinity of other pollution sources (e.g. wildfires) the background variability of CO becomes more complex and can interfere with with the retrieved local

emissions of Mexico City. Hence, a better model of the CO background concentration and its variability is needed to cope with this effect. However, Fig. 6 also shows that fitting the emissions of the different city districts is significantly improving the $\chi^2$ between TROPOMI and WRF-chem over the whole time range and the improvement can be even more than 50% for single overpasses compared to case of simulations only using the emission values provided by the INEM inventory.

For a correct interpretation of the retrieved emissions the averaging kernel, as shown in Fig. 7 for four example cases, offers

several advantages. The figure shows that generally the averaging kernels have high values on the diagonal indicating high

sensitivity to the quantity to be retrieved. It indicates that TROPOMI measurements can be used to distinguish emissions of the different urban districts of Mexico, with the exception of the emissions of district Tulancingo. Due to the small mean emission, the averaging kernel indicates a low sensitive of the data product. Furthermore, the averaging kernel shows cross-correlations between the different elements of the state vector due to the regularization. Although these interdependencies exist, e.g. between the emission of CdMx and ACdMx as shown in panel (d) of Fig. 7, these are still small compared the diagonal. The averaging kernel information is very useful to filter the emission product with respect to the information provided by the TROPOMI measurements. Using the sensitivity of individual sources, this results in different number of coincidences for the different districts (panel (c) of Fig 5). This form of data mining optimizes the data use, keeping in mind that TROPOMI overpasses may be appropriate to determine one specific source but not all sources simultaneously. In this manner, error propagation in the inversion can be minimized.

Due to the little scatter and the higher data amount of the final data product for the suburbs CdMx and ACdMx allows to conclude on the time dependent variability of emission. Figure 8 (a) shows the time series of the emission for CdMx and ACdMx, which vary around the priori value. This temporal variation is determined from the measurements as all prior information is assumed to be time invariant. The scatter of the data is still high and even includes negative values. Even though negative emissions are not physical we need to keep them in our analyzes because filtering negative noise can induce a positive bias in the mean. Panel (b) of the figure shows relatively high values of the diagonal elements of the averaging kernel for the emissions of the two urban districts. Finally, panel (c) of the figure indicates a clear weakly CO cycle in the data with low values during weekends. During the week, the CO emissions of the two districts do not differ significantly due to the error estimates and more TROPOMI data is required to further constrain the weekly cycle. We found that the CO values on Saturday and Sunday are equally low. An explanation for this could be that the main source of CO in Mexico City during the week is traffic which is responsible for the weekly cycle and the remaining sources like cooking, water heating, etc. should not change much during the weekend.

A similar weakly cycle is observed by Mexico City situ measurements provided by 29 SEDEMA ground stations. For each of the sites, we use data from 2017 to 2018 for the overpass time of TROPOMI (12h-15h local time), calculated a weakly cycle and group the data in the stations located in the CdMx urban area and those located in the wider area of the metropolis. Figure 9a depicts the median of all weakly cycles and the standard error of the mean with a clear minimum during weekends. The error bars indicate that the overall shape of the weekly cycles for the remaining days vary a lot from station to station.

The lower CO concentrations during the weekend are also detectable with column retrievals from ground-based FTIR measurements in Mexico City 2280 m.a.s.l 19.32°N and -99.18°E at the campus of the national University by the atmospheric science center (CCA). The used spectra are recorded in the mid infrared with a resolution of 0.075 cm$^{-1}$ (Bezanilla et al., 2014; Plaza-Medina et al., 2017) and the CO column and profile is retrieved using the standard NDACC retrieval strategy (García-Franco et al., 2018; Borsdorff et al., 2018a). Figure 9b shows the averaged weakly cycle with standard error derived from the column measurements. Due to the low data density at weekends we used the full-time range from the 5th December 2010 to the 10th September 2019 without filtering for the overpass time of TROPOMI. These independent ground-based measurements confirm the weekly CO cycle found in the TROPOMI data. In general, the TROPOMI CO data product agrees very

well with retrievals from ground-based FTIR measurements performed by the TCCON network world wide with an averaged bias less than $6 \pm 3.8$ppb and the bias with retrievals from NDACC measurements is even lower (Borsdorff et al., 2018a).

## 5   Conclusions

In this study, we analyzed TROPOMI CO retrieval from 551 overpasses of the instrument over Central Mexico, which cor-
responds to about 2-years of measurements starting from the 14 November 2017 until the 25 August 2019. We found that urban pollution can be monitored by the TROPOMI CO data. The high signal-to-noise ratio of the measurements allowed us to distinguish distinct CO enhancements over the various urban districts of Central Mexico using single orbit overpasses of TROPOMI with a high spatial resolution of 7x7 km$^2$ that is enhanced to 7x5.6 km$^2$ from the 6th of August 2019 onwards.

With a dedicated WRF-chem tracer simulation for the full-time range of the current TROPOMI data record, we could
distinguish the contribution of ten urban districts Tula, Pachuca, Tulancingo, Toluca, Cuernavaca, Cuautla, Tlaxcala, Puebla, CdMx, and ACdMx. The model data was collocated with the TROPOMI measurements and convolved with the total column averaging kernel to account for the vertical sensitivity of the instrument. Here, the WRF-chem tracer simulation does not account for atmospheric chemistry and so the simulated CO tracer fields is linear in the emission rates of the different districts. The model is extended by two effective parameters describing a spatially constant CO background and a dependency of the
simulated column on terrain height.

The CO emissions are determined in two steps. First, we apply a unregularized least squares fit of the model to the TROPOMI observations to determine the averaged emission per district. A strict data screening based on the measurements and WRF-chem model simulation reduced the TROPOMI data set from 551 to 199 overpasses. Second, we solve a regularized least squares problem, which minimizes the variation of the emission around its mean to reduce the noise propagation in the inversion. By
means of appropriate regularization parameters, we reduce the scatter of the retrieved emissions to about 60% of the median for all urban districts. For data interpretation and screening, the use of the averaging kernel is of great advantage. The final retrieval product includes a averaging kernel as a retrieval diagnostic, which allows to analyze retrieval sensitivities and cross correlations between the inferred emission rates.

The derived averaged emissions for the various urban districts of Mexico deviates significantly from emission estimates of
the "Inventario Nacional de Emissions de Contaminantes Criterio" (INEM) inventory adapted to the period 2017-2019. The TROPOMI emissions from the urban districts Tula ($0.10 \pm 0.004$ Tg/yr) and Pachuca ($0.09 \pm 0.005$ Tg/yr) in the Norther of Mexico City deviate significantly from the INEM inventory with 0.008 Tg/yr for both areas. For the emission of the central part of Mexico City (CdMx), TROPOMI indicate $0.14 \pm 0.006$ Tg/yr versus 0.25 Tg/yr INEM emissions and $0.28 \pm 0.01$ Tg/yr versus 0.14 Tg/yr INMEN emissions for the district ACdMx. Together, both districts have similar emissions with 0.42 Tg/yr
seen by TROPOMI versus 0.39 Tg/yr from the inventory, pointing to a different relative distribution of the CO emissions seen by TROPOMI. Moreover, using a posteriori data screening to optimize data selection per emission source allows us to distill a weakly cycle of CO emission at the districts CdMx and ACdMx from the data set with a clear minimum during weekends. This finding is in agreement with in-situ observations and ground-based FTIR measurement in the metropolis.

Our study shows the need and the potential of regional atmospheric transport modeling for the interpretation of TROPOMI CO data over metropolitan areas like Mexico City. Here, the CO pollution is a composite of emissions from different districts and its transport leads to complex CO enhancement patterns in the atmosphere. The WRF-chem tracer model could simulate the TROPOMI measurement to a great extent, however model errors are still significant and further improvement is required to fully explore the TROPOMI CO observations of urban sources. Another potential error source of our method are the accuracy of the week-to-week and monthly variations of the emissions in the INEM inventory considering the fixed overpass time of TROPOMI. Furthermore, basin cities can be problematic with low wind speed for days, which could lead to accumulate signals from more than one day in the basins which is not yet covered by our approach. To account for this effect in our inversion needs major adjustments, which will be investigated in follow up studies.

## 6 Data availability

The TROPOMI CO data set of this study is available for download at ftp://ftp.sron.nl/open-access-data-2/TROPOMI/tropomi/co/. The in-situ measurements in Mexico City were downloaded from http://www.aire.cdmx.gob.mx. The ground-based FTIR measurements in Mexico can be downloaded http://www.epr.atmosfera.unam.mx/ftir_data/UNAM/CO/VERTEX/v1/.

*Author contributions.* Tobias Borsdorff, and Jochen Landgraf performed the TROPOMI CO retrieval and data analysis. Agustin Garcia Reynoso, Gilberto Maldonado, and Bertha Mar-Morales performed the WRF-chem simulation. Wolfgang Stremme and Michel Grutter provided the ground-based FTIR measurements. All authors discussed the results and commented on the manuscript.

*Competing interests.* The authors declare no competing interests.

*Disclaimer.* The presented work has been performed in the frame of the Sentinel-5 Precursor Validation Team (S5PVT) or Level 1/Level 2 Product Working Group activities. Results are based on preliminary (not fully calibrated/validated) Sentinel-5 Precursor data that will still change. The results are based on S5P L1B version 1 data.

*Acknowledgements.* The presented material contains modified Copernicus data [2017,2018] The TROPOMI data processing was carried out on the Dutch national e-infrastructure with the support of the SURF Cooperative. This project was partially supported by LANCAD-UNAM-DGTIC-179 supercomputer system.

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

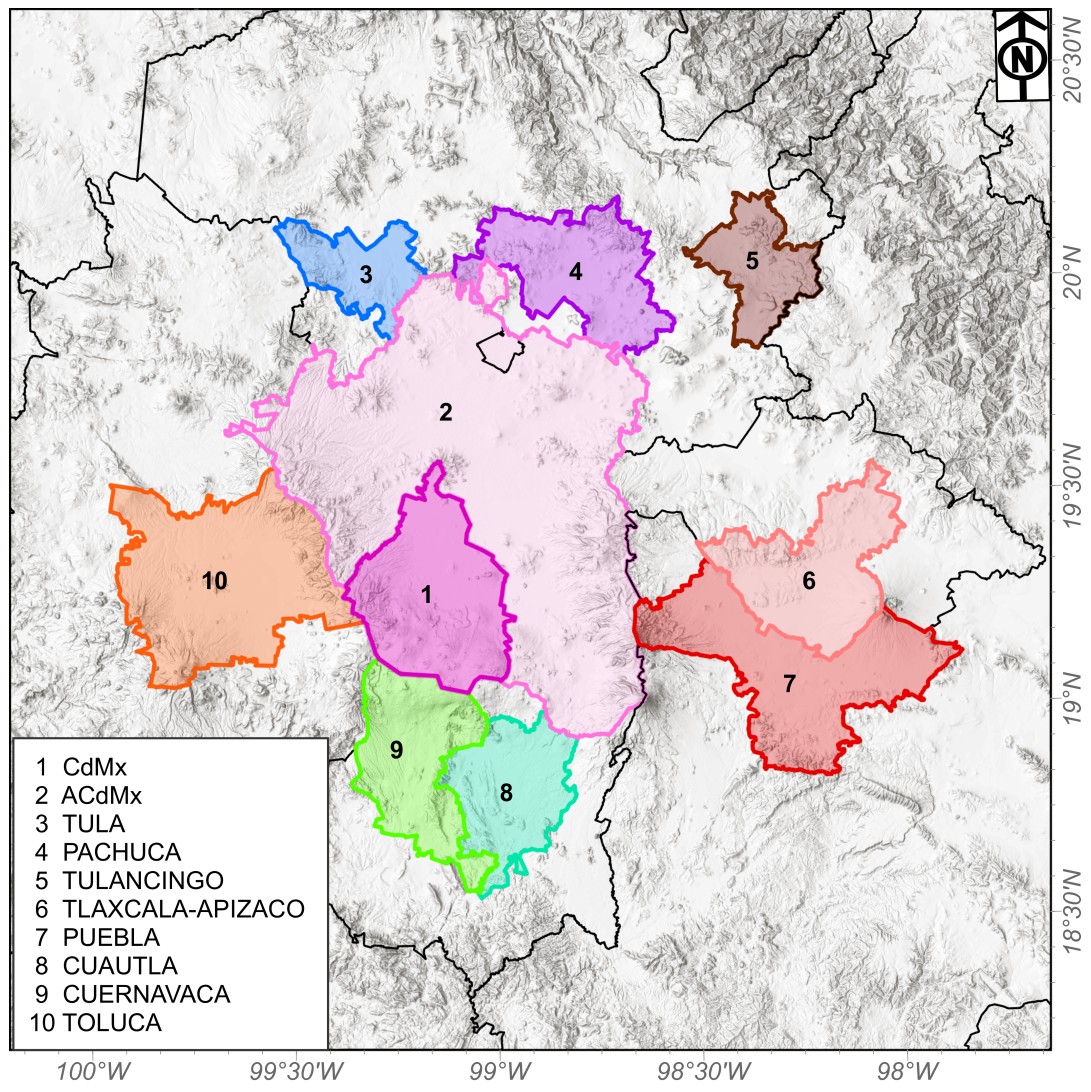

**Figure 1.** Urban districts surrounding Mexico City. For each of the color coded domains a separate WRF-chem tracer run was performed based on the emissions within the polygons. The elevation map in the background is under copyright © Esri, Airbus DS, USGS, NGA, NASA, CGIAR, N Robinson, NCEAS, NLS, OS, NMA, Geodatastyrelsen, Rijkswaterstaat, GSA, Geoland, FEMA, Intermap and the GIS user community.

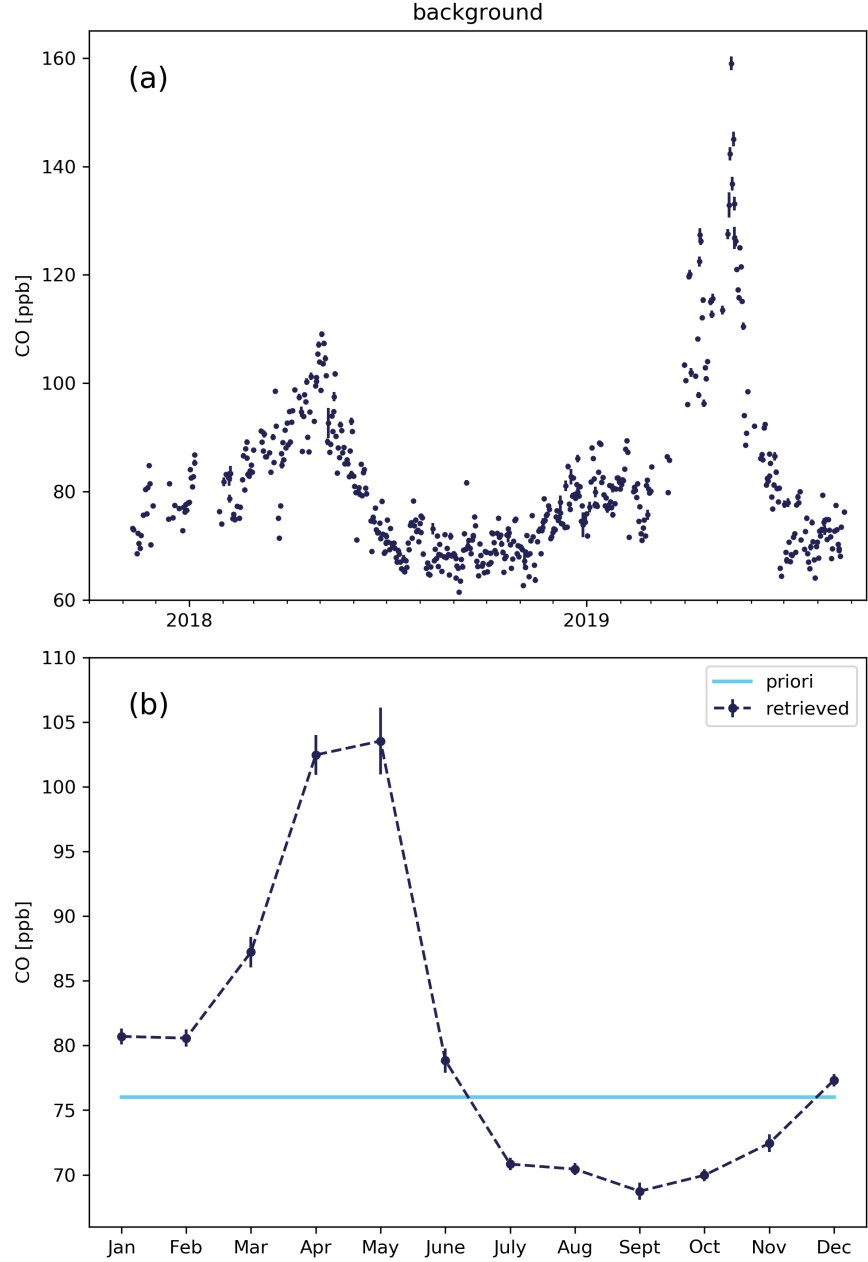

**Figure 2.** Background CO concentration for the domain shown in Fig. 1 estimated by fitting the WRF-chem simulation to the TROPOMI data. (a) background CO for individual collocations from the 9th of November 2017 to the 25th of August 2019. (b) Monthly mean background CO based on the individual collocations. The error bars are the standard error of the mean and the light blue line time invariant prior used in the fit.

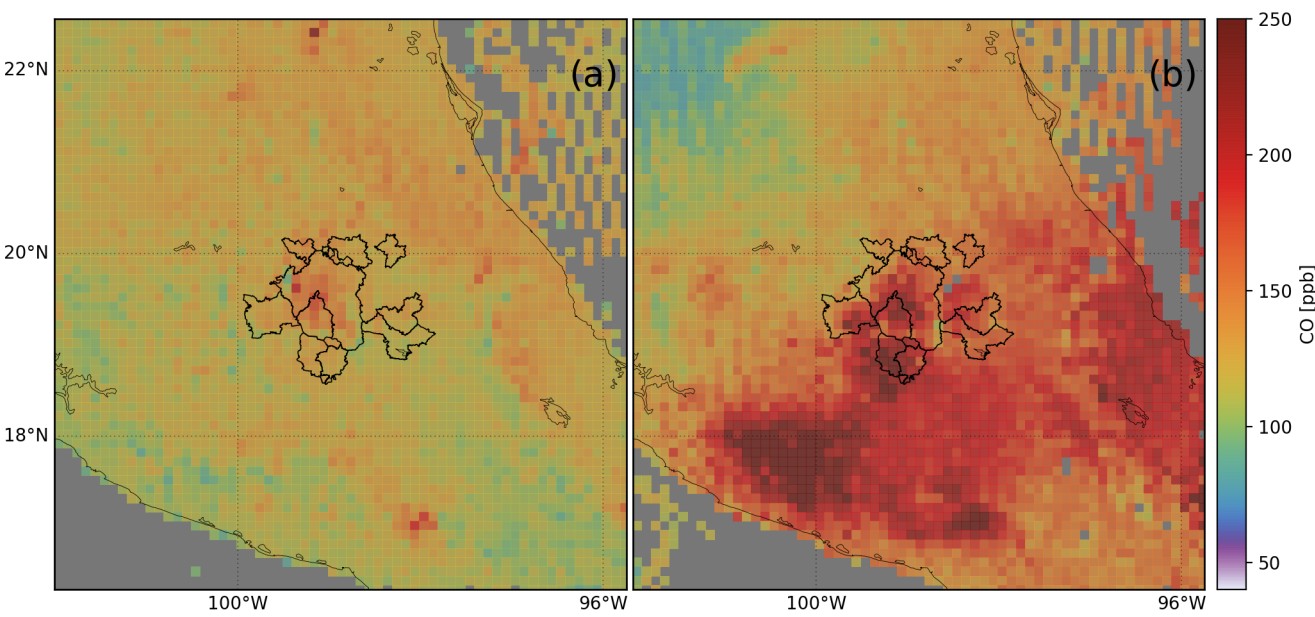

**Figure 3.** TROPOMI CO data over Mexico City averaged on a 0.1 by 0.1 degree lat/lon grid. (a) averaged from 12 to 18 of April 2019 showing undisturbed background CO levels. (b) averaged from 12 - 18 of May 2019 showing high CO concentrations in Mexico City caused by fires in the South-East. The street map in the background is under copyright © 2009 ESRI, AND, TANA, ESRI Japan, UNEP-WCMC.

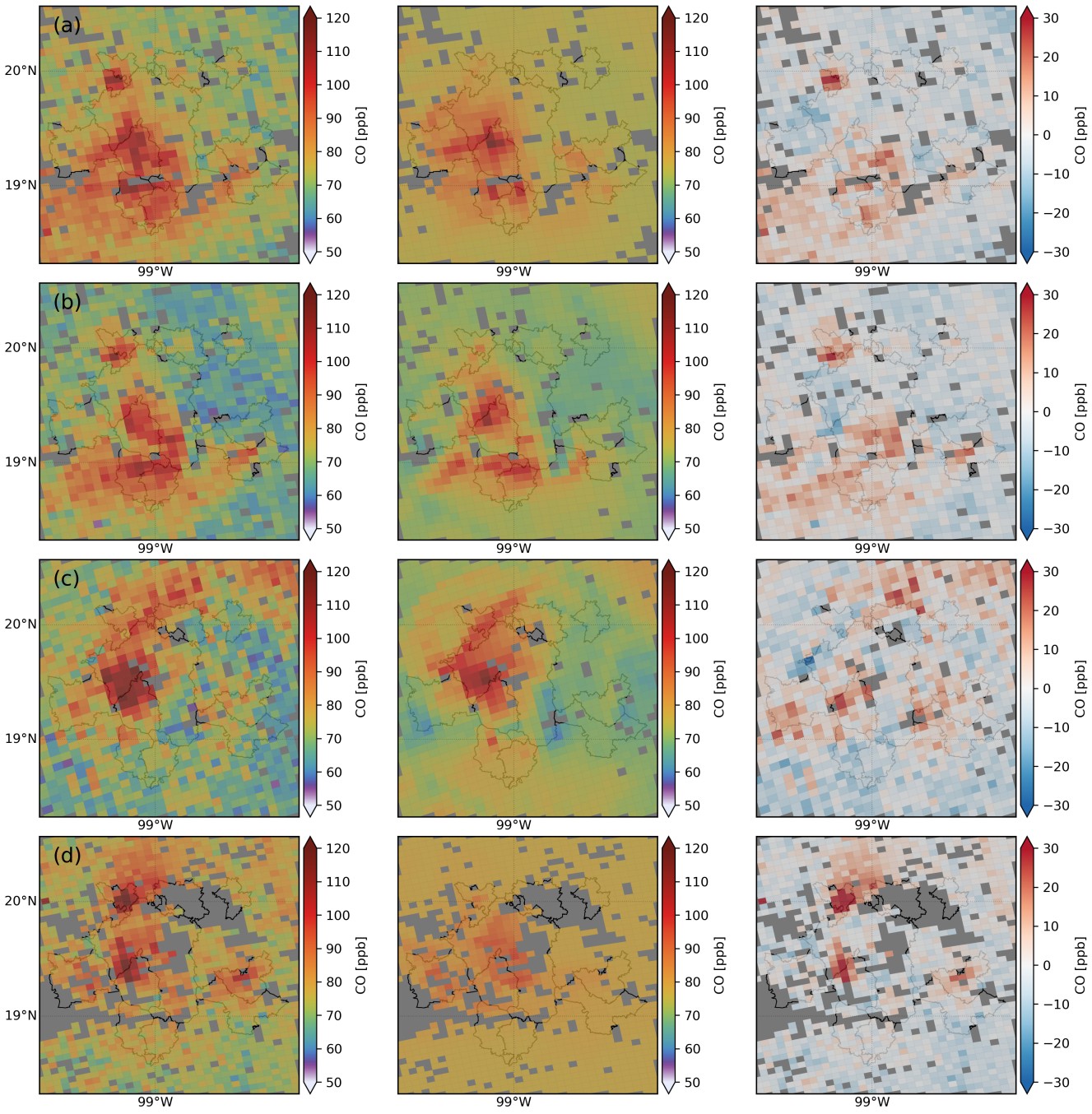

**Figure 4.** Example cases for fitting the WRF-chem simulation to the TROPOMI data deploying the "Final-fit" approach for (a) the 20th of September, (b) the 7th of November, (c) the 19th of November 2018 and (d) the 17th of August 2019. TROPOMI CO retrievals (left column), WRF-chem simulation fitted to the TROPOMI data (middle column), and the residual (right column, TROPOMI - WRF-chem).

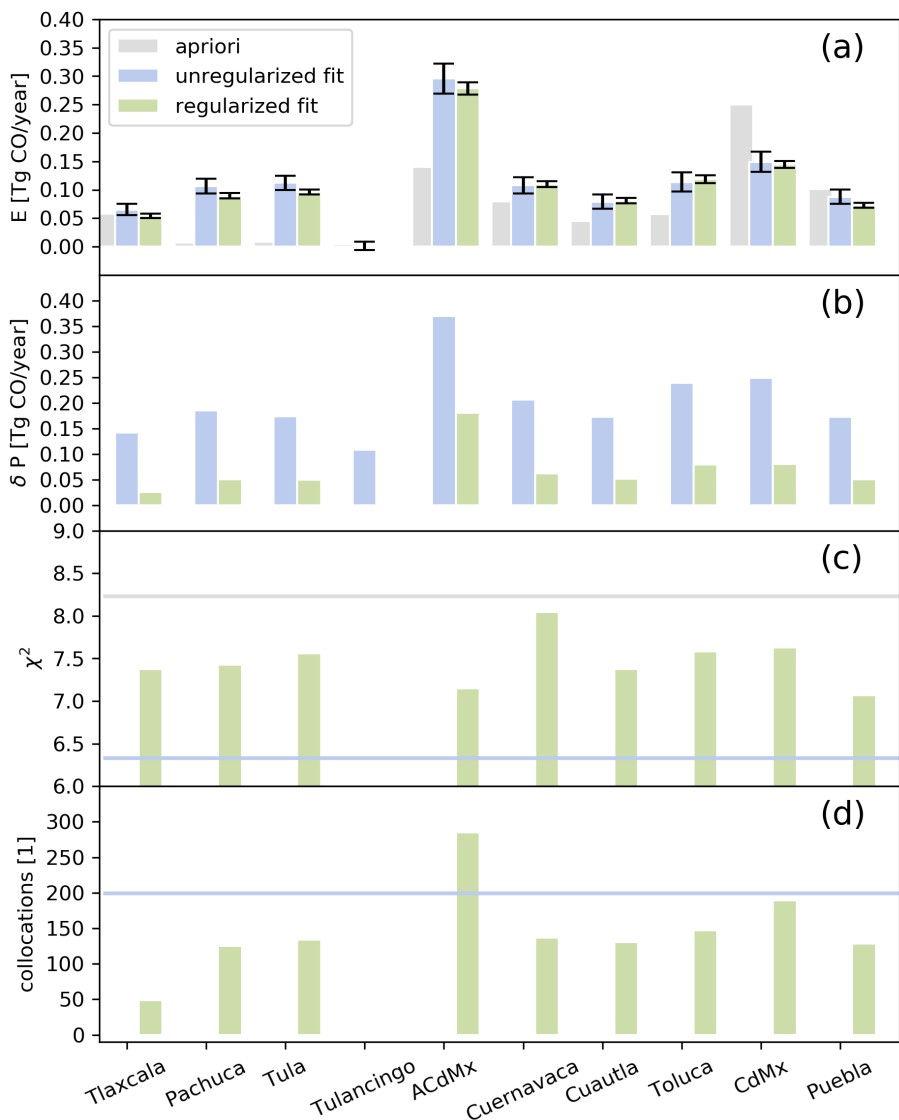

**Figure 5.** Statistics of CO emissions averaged from the 9th of November 2017 to the 25th of August 2019 for the tracer domains shown in Fig. 1. (a) Median of the priori emissions (adapted INEM inventory) used for the WRF-chem simulation (grey) and retrieved from the TROPOMI data (unregularized fit in blue, regularized fit in green). The error bars indicate the standard error of the mean calculated from the delta percentiles (b) used as a robust estimation of the standard deviation, (c) the median of the goodness of the fit ($\chi^2$), and the number of collocations (d). The number of collocations and the $\chi^2$ values of the apriori simulation and unregularized fit are the same for all tracer domains (blue and grey line) but in the final regularized fit it is changing due to the information content filtering. Here, a collocation corresponds to a specific day because TROPOMI overpasses the region only once.

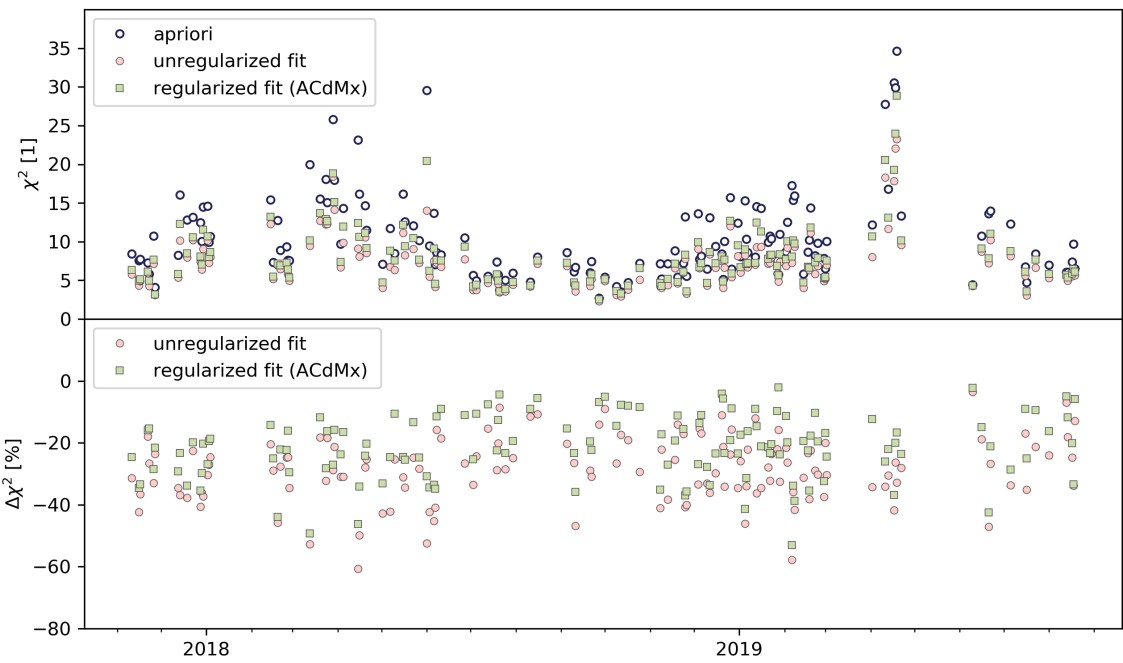

**Figure 6.** (a) The goodness of the fit ($\chi^2$) of the TROPOMI CO measurements and the simulation of the WRF-CHEM model for single orbit overpasses. We distinguish the cases; only fitting background parameters (apriori), and additionally fitting the 10 tracer fields (unregularized and regularized). (b) Improvement of the goodness of the fit ($\chi^2$) when fitting the 10 tracer fields (unregularized and regularized) relative to the (apriori) case. For the regularized fit we only show the ($\chi^2$) of the urban district ACdMx because it provides a good data coverage when filtering for the degree of freedom.

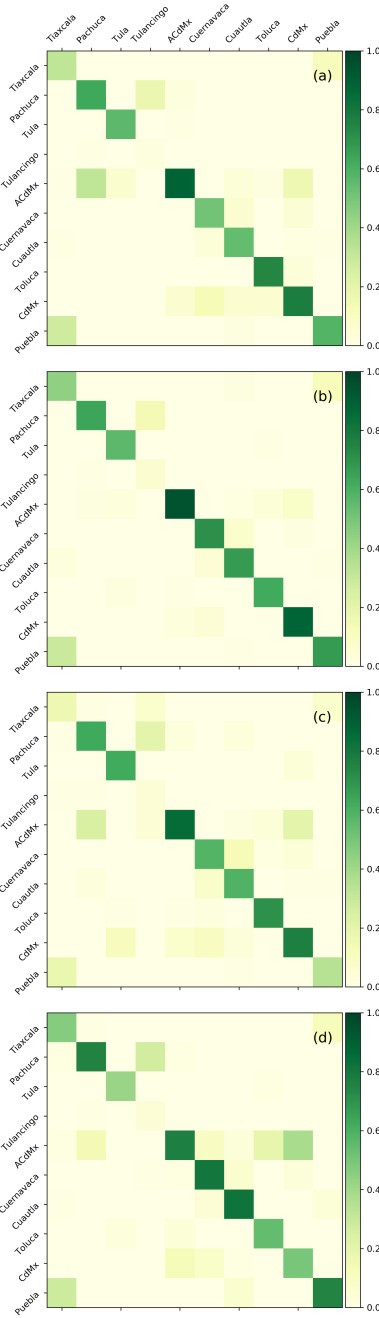

**Figure 7.** Averaging kernel matrices showing the sensitivity and cross-sensitivities for the scaling of the different tracer fields. The same cases as in Fig. 4 are shown for the dates (a) the 20th of September, (b) the 7th of November, (c) the 19th of November 2018 and (d) the 17th of August 2019 but deploying the regularized retrieval.

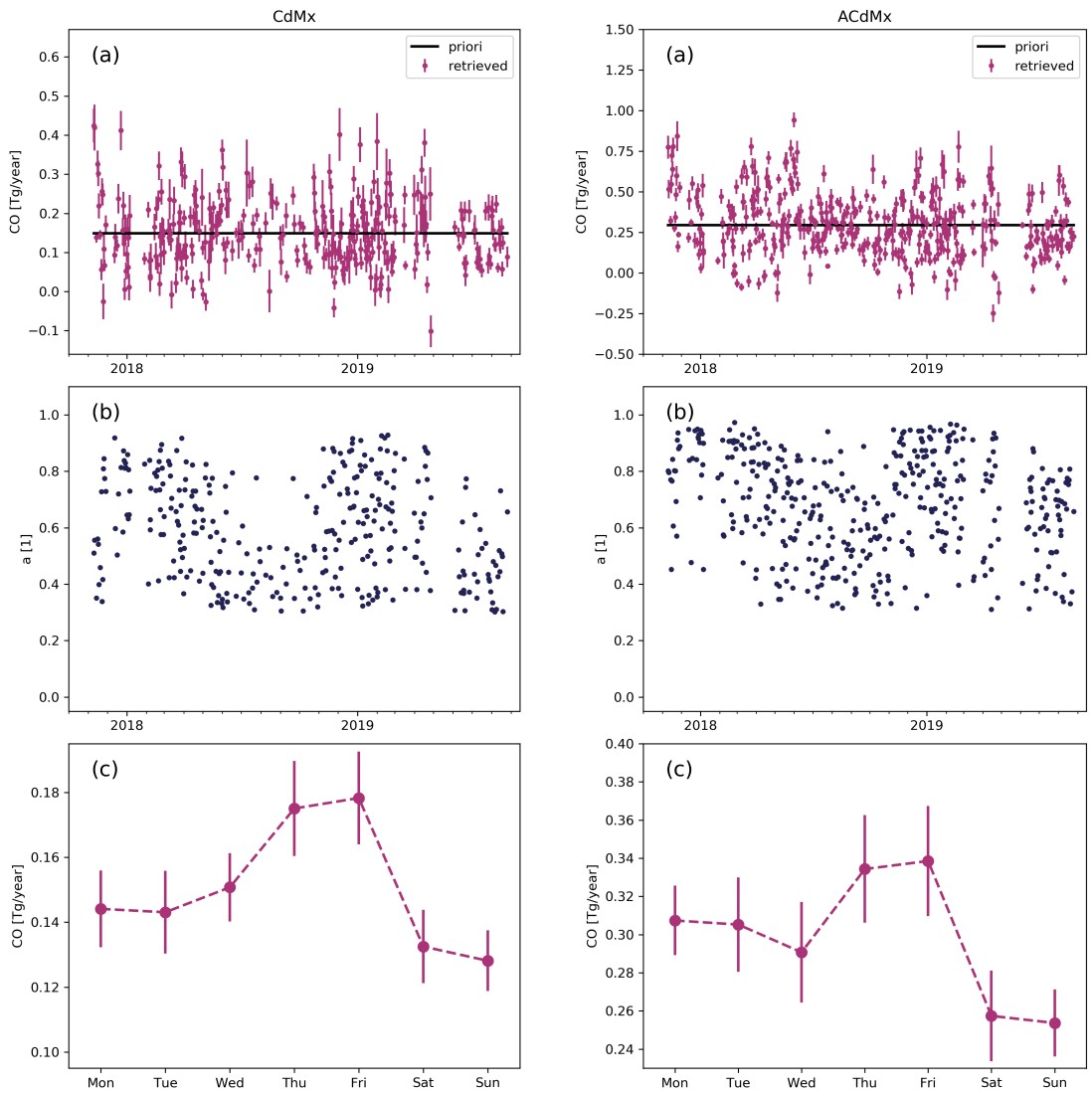

**Figure 8.** Retrieved CO Emissions from the TROPOMI data for the tracers CdMx (left panel) and ACdMx (right panel). (a) Time series of individual retrieved CO emissions. The error bars indicate the error of the fit and the black line is the time invariant priori used in the fit. (b) degree of freedom of the scaling factor for the tracer field. Only data with dofs > 0.3 is accounted for. (c) Weekly cycle of the CO emissions. Median values are shown and the error bars are the standard error of the mean deploying the delta percentile as a robust estimation of the standard deviation.

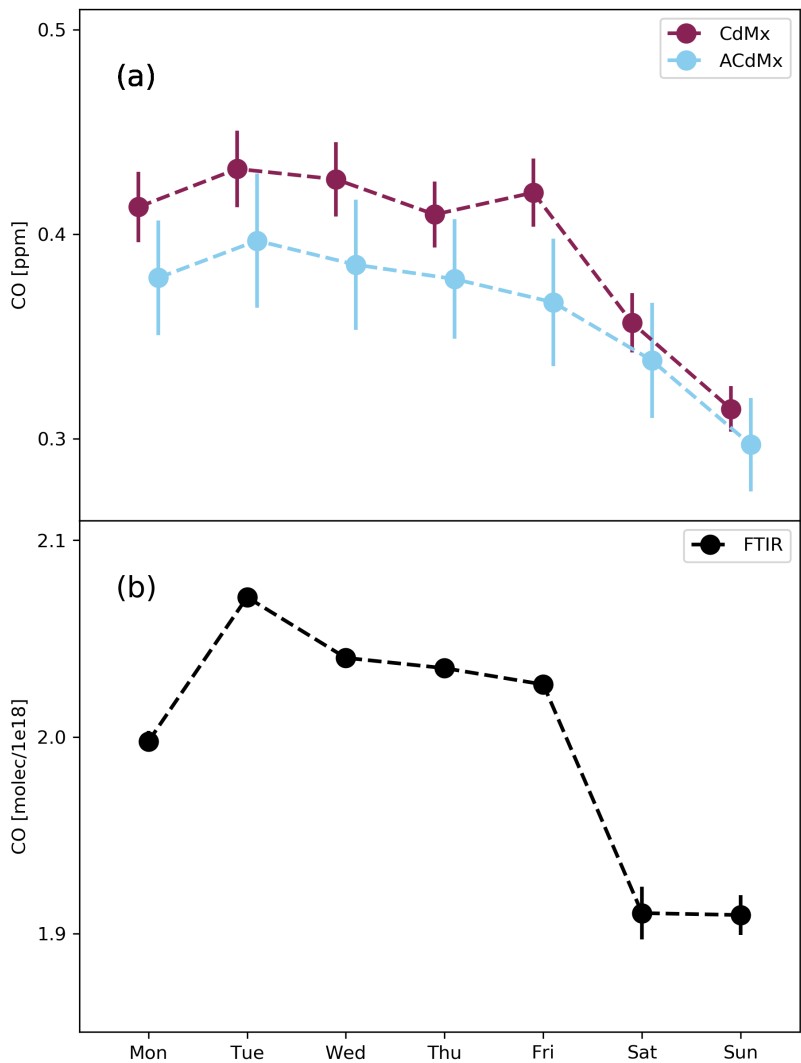

**Figure 9.** Weekly cycle of the CO concentration. (a) based on 29 in-situ measurements station operated by SEDEMA. (b) ground-based FTIRs vertical column measurements of an instrument located in Mexico. Median values are shown and the error bars are the standard error of the mean deploying the delta percentile as a robust estimation of the standard deviation.