# Peer review of "Monitoring CO emissions of the metropolis Mexico City using TROPOMI CO observations"

_Atmospheric Chemistry and Physics, 2020_

## Referee Comment (RC1) · Anonymous Referee #1 · 1 Jun 2020

Borsdorff et al. present an inversion approach to retrieve carbon monoxide (CO) emissions over ten urban districts of Central Mexico based on high-spatial resolution observations from the TROPOMI satellite and using WRF-Chem simulations. This work presents new insight into the needed high spatial distribution emissions using satellite observation and likely suitable for the journal. I do have some comments below that need to be addressed before publication.

Major Comments

One of the main concerns is regarding the CO background concentration and chemistry. Authors assume a time invariant CO background concentration, while I believe background processes in the region of interest and its surrounding are quite important. I highly suggest to describe in detail why a constant CO background CO has been used.

[Figure]

Please explain in detail how the background CO flowing into the domain produced by all non-metropolis Mexico City (10 districts) sources, including, non-metropolis Mexico City fires, is treated. Considering the relatively long lifetime of CO transport is extremely important.

Furthermore, biogenic non-methane VOCs emitted from vegetation might be important as a source for the chemical production of CO in the atmosphere. In the manuscript, I did not find information regarding these contribution, maybe it is too small for the metropolis?, what about the transport of the surroundings to the districts. It would be important to add a description on this.

Lastly, according to the authors the configuration of the model does not account for atmospheric chemistry, does that mean that Gas-phase Chemistry is not included?. Similarly, please include a description of why this configuration was chosen.

Specific Comments

Authors recognize the possible error sources, and if I understand correctly authors estimate uncertainties in the inversion, I highly suggest to include the uncertainties of emissions in the abstract.

P1, L2. It is mentioned that 551 overpasses are analyzed, please specify the exact time period. The season(s) might be relevant.

P1, L4. It is not clear to me if you use WRF coupled with Chemistry (WRF-Chem)?

P1, L8. Do you identify the sources missing in the INEM in Tula and Pachuca?

P1, 14. It is mentioned: "CDMX and ACDMX follow a clear weakly cycle with a minimum during the weekend" does this mean that the weekend effect is not found in the other regions?

Section2, TROPOMI CO data set:

In the current manuscript, I do not find a real value of including the FTIR observations,

however it might be good to include it in this section. I suggest to include comparisons between TROPOMI and FTIR for coincident dates, do they compare ok?

Section 3.1, The WRF model.

Important chemical parameterizations in the model are missing, e.g., what biogenic and biomass burning emissions are used?. What kind of boundary conditions?. Is the inflow of CO emitted by fires outside the region of interest included?. What time step is used?

P4, L17. Do you mean equation 1?

P5, L3-7. As in my major comment, it is a big assumption that " local enhancements of CO are due to emissions of the city districts of the same day" with a constant CO background?. It is well known that biomass/fire emissions can contribute significantly to the CO in the region. I wonder why an inflow of background CO is not taken into account, my understanding is that WRF-Chem can handle this.

In order to have a sense of the spatial distribution of CO, I highly suggest to include the urban districts in Fig. 3.

P8, L2-9. It is not clear how the background concentration was estimated.

P8, L9. It is mentioned that "the fire season many data cannot be considered", how many days (or percent) are excluded based on this?

Figure 4, It is hard to identify the districts on this figure, maybe you could include the contour/shapes of the districts.

Figure 5. I recommend to follow the names of the districts as in Figure 1. Especially for Ciudad de Mexico and CDMX.

Figure 5. Why does Tulancingo have a zero emission?

Figure 5. Is the number of collocations the same as the number of days?

**[ACPD](https://www.atmospheric-chemistry-and-physics.net/)**

Interactive
comment

Figure 7. What does negative CO emission mean?

Figure 7. It is hard to believe that emissions on Sat and Sunday are very similar, what time does it represent the emissions?

Figure 8. The weekly cycle of CO is considerably different than the weekly cycle of the emissions from Fig 7 (c), maybe I miss it but do you explain why?. Also, error bars from FTIR are extremely low, I do not think a standard deviation from the mean is the best way to characterize variability.

[Figure]

---

## Referee Comment (RC2) · Anonymous Referee #2 · 15 Jun 2020

General comments:

The paper presents the results of a two-step optimization using carbon monoxide column measurements from TROPOMI over Mexico City. The atmospheric model WRF coupled to a city inventory of CO emissions is used to generate response functions for 10 urban districts. About two years of data have been simulated to estimate the CO emissions for these districts. The simulations and the design of the response functions have been carefully constructed. This topic is highly relevant and the use of TROPOMI data to constrain the city inventory is sound. However, the study has two major problems that need to be addressed before publication:

- two-step minimization: The optimization of the emissions requires two steps. The first step (pre-fit) filters out a significant fraction of the TROPOMI data based on several

criteria, while the second step (final fit) includes the entire data set. Some of these criteria require more justification, especially when threshold values are bieng used without any justification. But more importantly, this two-step process implies that the same measurements have been used twice. This approach seems to be a solution to the noise affecting the model-data residuals, hence limiting the convergence of the optimization system. In figure 5, there is almost no difference between the pre-fit and the final fit. The results are already constrained after the "pre-fit" step, except that the error bars further decrease, which seems artificial without assimilating additional measurements.

To be clear, data should never be used twice in the optimization as it artificially increases the information content from the data set. Instead, the emissions should be produced by a single optimization procedure, iterative or at once, but extracting information only one time. If the noise is the inherent problem here, it should be treated by filtering out the noise in the data or in the model. Smoothing data signals, decomposing the signals into frequencies, or averaging over time (both model and data) will help extract the information from noisy model-data residuals. Other approaches like Wasserstein distance or other machine learning techniques will help remove the noise. The optimization procedure needs to be revised to produce robust emissions and uncertainties. As it stands, the selection of data is arbitrary and the optimization uses the sae data multiple times.

- Background determination: The determination of the background CO values is never explained in details. The first paragraph of Section 4 describes very briefly that CO background has been fitted. Which domain has been used? The entire State of Mexico? Considering the topography, the gradients over the domain, and the potential contamination by other sources (such as fires, but also cities, industries, shipping, ...), the uncertainty associated with the background determination needs to be quantified and included in the optimization. The uncertainty associated with the background seems to be equal to zero in the final fit (section 3.4). How is the uncertainty defined

in the optimization? Zero values appear in the prior error covariance matrix (P7-L14). It suggests that the background is pre-defined (before the final optimization). A section should describe precisely how the background values are determined in (or before) the optimization.

Assuming you provide a coherent one-step minimization procedure, and a robust determination of th background values for each day, I recommend that you include some pseudo-data experiments to evaluate the potential of your optimization to constrain the city emissions. Simple perturbations should be added to urban districts to determine the actual constraint from TROPOMI data.

- As a last comment, the selection of data with low wind speed conditions will increase the model errors. If the aboslute wind speed is 2 m/s, an error of 2 m/s in wind speed corresponds to an error of 100% on the emissions. It maximizes the local enhancements which helps with the large noise, but this filter is typically the opposite in most studies using satellite data (minimum wind speed). Removing noise will help removing that threshold which seems to be a simple but risky solution to reducing the noise in model-data residuals.

Technical comments:

P2 – L1: "and its transport in the atmosphere ": Unclear.

P2 – L20: Add references for the error sources

P2 – L30: "Here the emission estimation changed by 0.42 Tg/yr 30 in only 2 years from 2014 to 2016, due to a change in the mobile emission model from 'mobile' to 'moves'." This means that the emission model changed and not the emissions. Has this change been confirmed or validated by other data?

P4 – L5: "The inventory is time dependent and accounts for the diurnal, week-to-week and monthly variations of the emissions" How accurate are these cycles? Considering the overpass time is fixed, the mismatch can be explained by a difference in diurnal

cycles in and out of the city. How was the inventory constructed? Does it include traffic counts? Are the other sectors using temperature-dependent relationship?

P4 – L29: "Here, z is the mean elevation in the TROPOMI CO ground pixels and zref = 2240 m the reference altitude which is set to the elevation of Mexico City." This correction is unclear. The altitude used by TROPOMI is defined as a surface pressure. The altitude error depends on the difference between the WRF surface pressure and the TROPOMI surface pressure. Why using an average altitude of Mexico City as a reference?

P5 - L4: "local enhancements of CO are due to emissions of the city districts of the same day" Have you tested that assumption? Basin cities are often problematic with low wind speed for days, which can accumulate signals from more than one day in the basins (example: Los Angeles during Winter). An averaged wind speed or residence time of tracers would help justify this assumption.

P5: Use the current notation for multivariate regression used in most publications (observation operator H, state vector x, prior error cov B, Obs error cov R, observations y, Kalman gain K).

P6 - L17: "with little forward model errors". Unclear. Re-phrase.

P6 – L24: "for low wind speeds more reliable". The model errors are critical during low wind speed conditions when a slight change in wind speed can affect the magnitude of the observed enhancements. Typically, high wind speeds should be avoided because local enhancements are weak while low wind speeds should also be avoided when a small change in the wind speed can significantly change the local enhancements.

P6 – L30: "fit residuum â§Íδâ§Î' < 8ppb, and the standard deviation $\sigma(\delta)$ < 8ppb to limit the effect of too large forward model errors." TROPOMI is an averaged enhancement over a grid cell. Point sources will be under-estimated in the data as the plume will not be mixed over the entire grid cell. This bias has been presented by the TROPOMI

team. Can you conform the relationship between point sources over the domain and the location of these high model-data differences?

In addition, removing noisy pixels will artificially decrease the uncertainty by removing undesirable pixels. Some of these large model-data differences might be real transport errors or observation noise.

P7 – L1: "$\sigma(\delta)/\delta$(ymeas) < 0.65 to ensure that the forward model can explain the variability of the measured CO field." This value seems arbitrary. How did you define it?

P7 – L4: What I the impact on the seasonal distribution? Does it remove data evenly over the year? This filters are likely to bias your results over specific season. A figure showing the time depencen of the filtered data is needed (or statistics)

P7 – L15: "$\alpha$bg and $\alpha$elv are not regularized" How can you optimize the emissions without regularizing the background values? Are they pre-determined? How were they defined?

P7 – 15-20: The balance between prior information and data constraint is usually computed with the Chi2 normalized distance. A value near one will define the optimal balance between the two. Michalak, A., Hirsch, A., Bruhwiler, L., Gurney, K., Peters, W., and Tans, P.: Maximum likelihood estimation of co- variance parameters for Bayesian atmospheric trace gas sur- face flux inversions, J. Geophys. Res.-Atmos., 110, D24107, https://doi.org/10.1029/2005JD005970, 2005.

P7 – L21-24: This approach weighs toward pixels that are co-located with the sources. In other words, it selects preferentially the pixels above the city. In general, it should work but an evaluation perido would be helpful (with and without the filter) to measure the impact of the selection. This approach might bias the results if the model under-/over-estimate urban pixels.

P7 – L23: "temporal variation of the INEM emissions to be about 40% [. . .] vary with 60% around their average" How do you define the 60%? The link between 40% and

60% is not explained. In addition, temporal variations and mean emission errors are not supposed to scale together. This part needs to be described more carefully. The emission errors should depend on the emissions alone instead of their temporal variability.

Figure 4: This figure provides illustrations but is not very helpful to prove that the WRF model is reliable or good enough. Instead, model-data mismatches should be presented in a synthetic figure, for different times of year, using whisker boxes. Snapshots for four days out of 160 is too few to convince the readers. This figure should be replaced.

P8 – L18: "This clearly shows that regional models like WRF have a great potential for the interpretation and analysis of TROPOMI data." No, this does not demonstrate the model capabilities nor the ability fo the model to extract emissions. Re-phrase.

P8 – L21: "For atmospheric conditions under high wind speeds the WRF simulations can deviate more from the TROPOMI measurements as shown in Fig. 4 (c)." This single day is too limited to conclude anything. More statistics on windy days are needed to prove your point is valid here.

Figure 5: The modeled and observed XCO should be presented first, summarized for the days available before and after filtering. Do the residuals show a seasonality? The results of the optimization are difficult to interpret without the evaluation of the initial model results.

P9 – L1-6: Large model-data mismatches are expected from observations, model errors, and prior errors. If observations are being used twice (if I understood correctly), mismatches will decrease automatically. Optimization should never be performed a second time with the same data. Unless I misunderstood the approach (different data are being used between these two steps), only one step should be performed. Otherwise the constraint from the data is over-estimated.

P9 – L7: "Furthermore, non-uniform variation of the background CO concentration can be a additional reason for this scatter (as shown in Fig. 3)." How did you determine the background? How do you separate the contribution from the city emissions? Is it all performed within the inversion? If so, how do you define background uncertainties? Some additional tests should be performed. If you introduce a background in the bias, is your optimization system able to recover that bias?

Figure 6: statistics should be presented for the entire data set and not only for four days.

P9 – L13-14: "the averaging kernel shows that the Final-fit inversion is insensitive to deviations of the Tulancingo emission from the prior estimate. Whereas the Pre-fit inversion estimates very small emissions for this district, the subsequent regularization changes the emission only marginally." This is the direct consequence of performing an optimization with the same data twice. Some of the constraint has already been introduced in the emissions.

Specific comments: P6 – L15: "depends crucially" = highly depends on

---

## Author Comment (AC1) · 31 Aug 2020

**author comments on the manuscript "Monitoring CO emissions of the metropolis Mexico City using TROPOMI CO observations", reviewer 1**

We would like to thank the reviewer for the constructive comments that aided us to improve our manuscript. In this document we provide our replies to the reviewer's comments. The original comments made by the reviewer are numbered and typeset in italic and bold face font. Following every comment, we give our reply. Here line numbers, page numbers and figure numbers refer to the original version of the manuscript, if not stated differently. Additionally, the revised version of the manuscript is added.

**1 Major Comments**

1. ***One of the main concerns is regarding the CO background concentration and chem- istry. Authors assume a time invariant CO background concentration, while I believe background processes in the region of interest and its surrounding are quite important. I highly suggest to describe in detail why a constant CO background has been used. Please explain in detail how the background CO flowing into the domain produced by all non-metropolis Mexico City (10 districts) sources, including, non-metropolis Mexico City fires, is treated. Considering the relatively long lifetime of CO transport is extremely important.***

   **adjusted** This is a misunderstanding, we are not assuming a time invariant CO background concentration in our study. Figure 2a shows for each TROPOMI overpass of Mexico City which background CO concentration was used. The time series clearly reflects elevated background CO concentrations during dry season (e.g. fire contribution) and low background CO during rainy season over Mexico City. Actually, we found, that the CO background concentration is a crucial component of our emission inversion scheme and therefore, decided to retrieve it together with the CO emission of the 10 city districts of Mexico (parameters $\alpha_{bg}, \alpha_{elev}$ in Equation 6). This ensures that the inversion scheme has the capability to decided itself which part of the TROPOMI measurement to interpreted as background and which as contribution from the city districts. Hence, all contribution from other sources excluding the 10 city districts will be represented by the fitted background parameters.
   To make this clearer, we changed the definition of the forward model in section 3.1. Now the background parameters are exclusively introduced as effective fit parameters in Eq. 5. Furthermore, we changed the paragraph p5, l27 from:
   " Finally, to improve the capability of the forward model to fit TROPOMI observations, we induce a linear altitude dependence of the simulated CO column $\mathbf{k}_{elv} = z - z_{ref}$. Here, $z$ is the mean elevation in the TROPOMI CO ground pixels and $z_{ref} = 2240$ m the reference altitude which is set to the elevation of Mexico City.

$$\mathbf{F}_{sat}(E_1, \cdots, E_{10}, \alpha_{bg}) = \sum_{i=1}^{10} \mathcal{O}(\tilde{\mathbf{k}}_i)E_i + \mathbf{k}_{bg}\alpha_{bg} + \mathbf{k}_{elv}\alpha_{elv} \tag{5}$$

   With these additional degrees of freedom the forward model can mitigate shortcomings of the WRF simulations using a spatially constant CO background. "
   to
   " To improve the capability of our forward model to fit TROPOMI observations, we introduce a spatially constant CO background field $\mathbf{k}_{bg}$ and an altitude dependence term $\mathbf{k}_{elv} = z - z_{ref}$ with corresponding scaling factors $\alpha_{bg}$ and $\alpha_{elv}$. Here, $z$ is the respective elevation of the TROPOMI CO ground pixels and $z_{ref} = 2240$ m is an arbitrary reference altitude set to the elevation of Mexico City,

$$\mathrm{F}_{sat}(E_1, \cdots, E_{10}, \alpha_{bg}, \alpha_{elv}) = \sum_{i=1}^{10} \mathcal{O}(\tilde{\mathbf{k}}_i)E_i + \mathbf{k}_{bg}\alpha_{bg} + \mathbf{k}_{elv}\alpha_{elv} \ . \tag{5}$$

   "

2. ***Furthermore, biogenic non-methane VOCs emitted from vegetation might be important as a source for the chemical production of CO in the atmosphere. In the manuscript, I did not find information regarding these contribution, maybe it is too small for the metropolis?, what about the transport of the surroundings to the districts. It would be important to add a description on this.***

**adjusted** The WRF model in this study is run in transport only mode. Hence, the chemical production of CO is not accounted for. However, it should be small compared to the other sources. All type of contributions like this (biogenic) or from outside the domain such as fires, power plants, other cities as well as contribution from global CO is compensated by the fitted background parameters $\alpha_{bg}, \alpha_{elev}$. Hence, when the background CO field becomes too complex or inhomogeneous e.g. as discussed for the CO from wild fires in Fig. 3, our approach will fail to reproduce the TROPOMI measurements and these cases are rejected.

We changed the sentence p4,l14 from:

" Further, the forward model assumes linear dependence of CO background field $\mathbf{k}_{bg}$ with scaling parameter $\alpha_{bg}$ ..."

to

" These two effective model components account for CO contribution over the Mexico City area originating from outside the model domain such as fires, power plants, biogenic production, other cities as well as the long-range transport (Borsdorff et al., 2019) and an altitude dependent linear vertical gradient of the CO columns. Both do not interfere with any localized emission sources. They mitigate shortcomings of the WRF-chem simulations ignoring CO boundary conditions at the model domain. "

3. *Lastly, according to the authors the configuration of the model does not account for atmospheric chemistry, does that mean that Gas-phase Chemistry is not included?. Similarly, please include a description of why this configuration was chosen.*

   **adjusted**

   We changed the sentence p3,l29 from:

   " ...and does not account for atmospheric chemistry (Dekker et al., 2017)."

   to

   " We ignore photo-chemical oxidation and secondary production of CO in the atmosphere (chem_opt option 106 (RADM2-KPP), as a tracer with gaschem off), which is justified by the long lifetime of CO compared with the size of the model domain as discussed by Dekker et al. (2017). Especially, for the region of Mexico City the contribution of atmospheric chemistry to the total CO concentration is less than 3% as presented by Mejia (2020). Hence, WRF-chem simulates the transport of CO surface emission as traces as done by e.g. Borsdorff et al. (2019), Dekker et al. (2017, 2018). "

**2    Specific Comments**

1. *Authors recognize the possible error sources, and if I understand correctly authors estimate uncertainties in the inversion, I highly suggest to include the uncertainties of emissions in the abstract.*

   **adjusted**

   We changed the sentence p1,l8 from:

   " ...0.10 Tg/yr and 0.08 Tg/yr CO"

   to

   " 0.10 ± 0.004 Tg/yr and 0.09 ± 0.005 Tg/yr CO "

   We changed the sentence p1,l10 from:

   " For CDMX, TROPOMI estimates emissions of 0.14 Tg/yr ..."

   to

   " On the other hand for Ciudad de Mexico, TROPOMI estimates emissions of 0.14 ± 0.006 Tg/yr CO, ..."

   We changed the sentence p1,l11 from:

   " ACDMX area, however, has a higher emissions with 0.29 Tg/yr according to TROPOMI observations ..."

   to

   " ...Arena Ciudad de Mexico the emission is 0.28 ± 0.01 Tg/yr according to TROPOMI observations ..."

   We changed the sentence p1,l10 from:

   " ...(0.43 Tg/yr TROPOMI versus 0.39 Tg/yr adapted INEM emissions). "

   to

   " ...(0.42 ± 0.016 Tg/yr TROPOMI versus 0.39 Tg/yr adapted INEM emissions)."

In addition we changed the same statements in the results and conclusion section p8,l30-43.

2. **P1, L2. It is mentioned that 551 overpasses are analyzed, please specify the exact time period. The season(s) might be relevant.**

   **adjusted**

   We changed the sentence p1,l2 from:
   " . . . (more than 2 years of measurements) using . . . "
   to
   ". . . we analyze TROPOMI observations over Mexico City in the period 14 November 2017 to 25 August 2019 by . . . "

3. **P1, L4. It is not clear to me if you use WRF coupled with Chemistry (WRF-Chem)?**

   **adjusted**

   We changed the sentence p1,l4 from:
   " . . . regional Weather Research and Forecasting (WRF) model to conclude . . . "
   to
   " . . . regional Weather Research and Forecasting (WRF-chem) model to conclude . . . "

   Accordingly, we changed the Acronym in the whole document

4. **P1, L8. Do you identify the sources missing in the INEM in Tula and Pachuca?**

   **adjusted**

   We added the following sentence p8,l30:
   " It is not yet clear what sources are missing in the inventory, this needs to be addressed in future studies. However, we identified an oil refinery and a power plant near to Tula and cement and lime kilns near to Pachuca that could contribute to the CO emissions. "

5. **P1, l4. It is mentioned: "CDMX and ACDMX follow a clear weakly cycle with a minimum during the weekend" does this mean that the weekend effect is not found in the other regions?**

   **not adjusted**

   No, but we cannot not conclude on it yet. We need to wait for more TROPOMI data to analyze the remaining districts.

6. **Section2, TROPOMI CO data set: In the current manuscript, I do not find a real value of including the FTIR observations, however it might be good to include it in this section. I suggest to include comparisons between TROPOMI and FTIR for coincident dates, do they compare ok?**

   **not adjusted** The agreement between TROPOMI and the FTIR measurements is already analyzed in Borsdorff et al. (2018). We found in general a good agreement with a low bias. The FTIR measurements show that the weekly cycle in CO can be detected in the total column concentration and by that adds extra information to weekly cycle that is detected by in-situ measurements at the surface. Hence, we would like to keep the FTIR measurements here.

7. **Section 3.1, The WRF model. Important chemical parameterizations in the model are missing, e.g., what biogenic and biomass burning emissions are used?. What kind of boundary conditions?. Is the inflow of CO emitted by fires outside the region of interest included?. What time step is used?**

   **adjusted**

   We added the following sentence p3,l29:
   " . . . (chem_opt option 106 (RADM2-KPP), as a tracer with gaschem off) . . . "

   Please also see major comment 3 of referee 1.

8. *P4, L17. Do you mean equation 1?*

   **corrected**

9. *P5, L3-7. As in my major comment, it is a big assumption that local enhancements of CO are due to emissions of the city districts of the same day with a constant CO background?. It is well known that biomass/fire emissions can contribute significantly to the CO in the region. I wonder why an inflow of background CO is not taken into account, my understanding is that WRF-Chem can handle this.*

   **adjusted**

   Please also see our answer to the major comment of the referee. We changed the paragraph p5,l3-7 from:
   " In our simulation of TROPOMI CO observations, we assume that the local enhancements of CO are due to emissions of the city districts of the same day, whereas emissions from outside the domain as well as the temporal accumulation of CO emission of the domain is described by the background CO field. Therefore, it means that the inferred emissions $E_i$ represents an emission estimate of the urban district for the particular observation day. Moreover, the effective model parameter $\alpha_{bg}$ and $\alpha_{elv}$ may vary between different TROPOMI overpasses.
   to
   " Finally, for the interpretation of our CO forward simulations, we make an important assumption. Although the WRF simulations account for the temporal accumulation of the localized CO emission over days and weeks, we allocate an emission estimate of the corresponding overpass time to each TROPOMI overpass. Here, we assume that a TROPOMI CO image is dominated by the emissions of the urban districts for the particular observation day, where the temporal accumulation of CO from previous days is partly described by the WRF simulation due to the corresponding scaling of the inventory and partly mitigated by fitting the nuisance parameter $\alpha_{bg}$ and $\alpha_{elv}$. "

10. *In order to have a sense of the spatial distribution of CO, I highly suggest to include the urban districts in Fig. 3.*

    **adjusted** The figure is updated as suggested.

11. *P8, L2-9. It is not clear how the background concentration was estimated.*

    **adjusted**

    We changed the sentence p8,l1 from:
    " Fig. 2 shows the fitted CO background concentration and its annual cycle."
    to
    " Fig. 2 shows the CO background that was fitted as an auxiliary parameter during the inversion described in Sec. 3.2. The concentration and its annual cycle is shown. "

12. *P8, L9. It is mentioned that the fire season many data cannot be considered, how many days (or percent) are excluded based on this?*

    **adjusted**

    We changed the sentence p8,l9 from:
    " Only fitting a scaling to a constant background field is not sufficient in this extreme cases and so during the fire season many data cannot be considered. . . . "
    to
    " Only fitting a scaling to a constant background field is not sufficient in these extreme cases and so during the fire season many data cannot be considered (we excluded the month May and June 2019). "

13. *Figure 4, It is hard to identify the districts on this figure, maybe you could include the contour/shapes of the districts.*

    **adjusted**

    The figure is updated as suggested.

14. ***Figure 5. I recommend to follow the names of the districts as in Figure 1. Especially for Ciudad de Mexico and CDMX.***

   **adjusted**

   We updated Fig. 5,6,7, and 8 as well as the whole text of the Manuscript. The term "Ciudad de Mexico" is replaced by ACdMx and "CDMX" by "CdMx". Hence, we are now following the nomenclature shown in Figure 1.

15. ***Figure 5. why does Tulancingo have a zero emission?***

   **adjusted**

   We changed the sentence p9,l11- from:
   " Moreover, the averaging kernel shows that the Final-fit inversion is insensitive to deviations of the Tulancingo emission from the prior estimate. Whereas the Pre-fit inversion estimates very small emissions for this district, the subsequent regularization changes the emission only marginally. "
   to
   " The figure shows that generally the averaging kernels have high values on the diagonal indicating high sensitivity to the quantity to be retrieved. It indicates that TROPOMI measurements can be used to distinguish emissions of the different urban districts of Mexico, with the exception of the emissions of district Tulancingo. Due to the small mean emission, the averaging kernel indicates a low sensitive of the data product. "

16. ***Figure 5. Is the number of collocations the same as the number of days?***

   **adjusted**

   We added the following sentence to Figure 5:
   " Here, a collocation corresponds to a specific day because TROPOMI overpasses the region only once. "

17. ***Figure 7. what does negative CO emission mean?***

   **adjusted**

   We added the following sentence p9,l25:
   " The scatter of the data is still high and even includes negative values. Even though negative emissions are not physical we need to keep them in our analyzes because filtering negative noise can induce a positive bias in the mean. "

18. ***Figure 7. It is hard to believe that emissions on Sat and Sunday are very similar, what time does it represent the emissions?***

   **adjusted**

   We added the following sentence p9,l29:
   " We found that the CO values on Saturday and Sunday are equally low. An explanation for this could be that the main source of CO in Mexico City during the week is traffic which is responsible for the weekly cycle and the remaining sources like cooking, water heating, etc. should not change much during the weekend. "

19. ***Figure 8. The weekly cycle of CO is considerably different than the weekly cycle of the emissions from Fig 7 (c), maybe I miss it but do you explain why?. Also, error bars from FTIR are extremely low, I do not think a standard deviation from the mean is the best way to characterize variability.***

   **not adjusted**

   We discussed this point on p9, l34. The variability of the weekly cycle is to high to conclude on its form yet. This will be revisited when we have more TROPOMI CO data available.

**References**

Borsdorff, T., aan de Brugh, J., Hu, H., Hasekamp, O., Sussmann, R., Rettinger, M., Hase, F., Gross, J., Schneider, M., Garcia, O., Stremme, W., Grutter, M., Feist, D. G., Arnold, S. G., De Mazière, M., Kumar Sha, M., Pollard, D. F., Kiel, M., Roehl, C., Wennberg, P. O., Toon, G. C., and Landgraf, J.: Mapping carbon monoxide pollution from space down to city scales with daily global coverage, Atmospheric Measurement Techniques Discussions, 2018, 1–19, https://doi.org/10.5194/amt-2018-132, URL https://www.atmos-meas-tech-discuss.net/amt-2018-132/, 2018.

Borsdorff, T., aan de Brugh, J., Pandey, S., Hasekamp, O., Aben, I., Houweling, S., and Landgraf, J.: Carbon monoxide air pollution on sub-city scales and along arterial roads detected by the Tropospheric Monitoring Instrument, Atmospheric Chemistry and Physics, 19, 3579–3588, https://doi.org/10.5194/acp-19-3579-2019, URL https://www.atmos-chem-phys.net/19/3579/2019/, 2019.

Dekker, I. N., Houweling, S., Aben, I., Röckmann, T., Krol, M., Martínez-Alonso, S., Deeter, M. N., and Worden, H. M.: Quantification of CO emissions from the city of Madrid using MOPITT satellite retrievals and WRF simulations, Atmospheric Chemistry and Physics, 17, 14 675–14 694, https://doi.org/10.5194/acp-17-14675-2017, URL http://dx.doi.org/10.5194/acp-17-14675-2017, 2017.

Dekker, I. N., Houweling, S., Pandey, S., Krol, M., Röckmann, T., Borsdorff, T., Landgraf, J., and Aben, I.: The origin of CO sources during the 2017 high pollution episode in India determined with TROPOMI and WRF data, manuscript in prep., 2018.

Mejia, J. F.: Running WRF in an Atmospheric Modeling Class: challenges and learning experiences, Atmosfera, 2020.

---

## Author Comment (AC2) · 31 Aug 2020

**author comments on the manuscript "Monitoring CO emissions of the metropolis Mexico City using TROPOMI CO observations", reviewer 2**

We would like to thank the reviewer for the constructive comments that aided us to improve our manuscript. In this document we provide our replies to the reviewer's comments. The original comments made by the reviewer are numbered and typeset in italic and bold face font. Following every comment we give our reply. Here line numbers, page numbers and figure numbers refer to the original version of the manuscript, if not stated differently. Additionally, the revised version of the manuscript is added.

**1 General Comments**

1. ***two-step minimization: The optimization of the emissions requires two steps. The first step (pre-fit) filters out a significant fraction of the TROPOMI data based on several criteria, while the second step (final fit) includes the entire data set. Some of these criteria require more justification, especially when threshold values are bieng used without any justification.***

   **adjusted**

   We relaxed the filter criteria for the Pre-fit retrieval and by that were removing the criteria criticized by the referee. The changes in our results are insignificant (please see the attached Fig.1 for a comparison). Accordingly, we recalculated everything and changed all the resulting numbers in the manuscript. The description of the filter criteria (p6,l23- p7l5) is changed from:
   " (3) The quality of the forward model depends on the meteorological situation, where we consider model simulations for low wind speeds more reliable. This considerations led to the criteria of the data filtering for the pre-fit. Thus, we only select overpasses which meet all of following filter criteria:

   - 70 % of the data domain is covered by TROPOMI observations
   - for all observations the across track pixel size is < 15 km.
   - the average wind speed of the scene is < 4 m/s.
   - The fit residuum $\langle \delta \rangle$ < 8ppb, and the standard deviation $\sigma(\delta)$ < 8ppb to limit the effect of too large forward model errors.
   - $\sigma(\delta)/\delta(\mathbf{y}_{\mathrm{meas}})$ < 0.65 to ensure that the forward model can explain the variability of the measured CO field.
   - $\sigma(\mathbf{F}(\mathbf{x}))$ > 4 ppb to ensure that the model data contain a clear pollution signatures.
   - the Pearson correlation coefficient $r$ > 0.3 between $CO_{TROPOMI}$ and $CO_{WRF}$.

   The filter criteria reduce the original set of 551 overpasses to 148, which we consider to be sufficient to estimate the overall average emission rate per district, yielding the prior state vector $\mathbf{x}_a$. "
   to
   " These considerations led to the criteria of the data filtering to determine the mean emission for each district. We only select overpasses which meet both filter criteria:

   - 70 % of the data domain is covered by TROPOMI observations
   - for all observations the across track pixel size is < 15 km.

   The filter criteria reduce the original set of 551 overpasses to 199, which we consider to be sufficient to estimate the overall average emission rate per district, yielding $\bar{x}$. "

   The statement of the referee about the optimization is addressed in the following comment.

2. ***But more importantly, this two-step process implies that the same measurements have been used twice. This approach seems to be a solution to the noise affecting the model-data residuals, hence limiting the convergence of the optimization system. In figure 5, there is almost no difference between the pre-fit and the final fit. The results are already constrained after the pre-fit step, except that the error bars further decrease, which seems artificial without assimilating additional measurements. To be clear, data should never be used twice in the optimization as it artificially in- creases the information content from the data set. Instead, the emissions should be produced by a single optimization procedure, iterative or at once, but extracting infor- mation only one time. If the noise is the inherent problem here, it***

*should be treated by filtering out the noise in the data or in the model. Smoothing data signals, decom- posing the signals into frequencies, or averaging over time (both model and data) will help extract the information from noisy model-data residuals. Other approaches like Wasserstein distance or other machine learning techniques will help remove the noise. The optimization procedure needs to be revised to produce robust emissions and un- certainties. As it stands, the selection of data is arbitrary and the optimization uses the sae data multiple times.*

**adjusted**

We completely rewrote section 3.2 about the inversion methodology (please see the new version of the manuscript). Our aim is to show that our two-step inversion scheme is extracting orthogonal information from the measurements and by that not the same information is extracted twice. "

3. *Background determination: The determination of the background CO values is never ex- plained in details. The first paragraph of Section 4 describes very briefly that CO background has been fitted. Which domain has been used? The entire State of Mex- ico? Considering the topography, the gradients over the domain, and the potential contamination by other sources (such as fires, but also cities, industries, shipping, . . . ), the uncertainty associated with the background determination needs to be quanti- fied and included in the optimization. The uncertainty associated with the background seems to be equal to zero in the final fit (sec- tion 3.4). How is the uncertainty defined in the optimization? Zero values appear in the prior error covariance matrix (P7-L14). It suggests that the background is pre-defined (be- fore the final optimization). A section should describe precisely how the background values are determined in (or before) the optimization. Assuming you provide a coherent one-step minimization procedure, and a robust de- termination of th background values for each day, I recommend that you include some pseudo-data experiments to evaluate the potential of your optimization to constrain the city emissions. Simple perturbations should be added to urban districts to determine the actual constraint from TROPOMI data.*

**adjusted**

In our approach we chose to infer information about the background CO directly from the TROPOMI CO measurements by simultaneously fitting the parameters $(\alpha_{\mathrm{bg}}, \alpha_{\mathrm{bg}})$ of Eq. 5) together with the emission estimates of the different city districts in the same domain. This is advantageous because the inversion can decide itself which part of the measured signal is classified as background and which as pollution from the city districts which minimized the change to introduces biases in the emission estimates. For the same reason we are fitting the background parameters always without imposing a regularization for each TROPOMI orbit. Here, the parameter $\alpha_{\mathrm{bg}}$ models a background CO concentration and $\alpha_{\mathrm{bg}}$ a gradient of the vertical CO concentration dependent on the orography of the region.
Please see our answer to the third major comment of referee 1 and the corresponding changes to the manuscript.
The zeros in the regularization matrix ensure that the background parameters are not regularized. To make this more clear We change the sentence p7,l14:
from
" such that the elements of the state vector $\alpha_{\mathrm{b}g}$ and $\alpha_{elv}$ are not regularized. "
to
" . . . where the zeros ensure that the elements of the state vector $\alpha_{\mathrm{bg}}$ and $\alpha_{\mathrm{elv}}$ are not regularized. Obviously, the . . . "
Furthermore, the errors of the background are shown in Fig.2 for each point. They are small because we have a strong signal about the background in the TROPOMI data.

4. *As a last comment, the selection of data with low wind speed conditions will increase the model errors. If the aboslute wind speed is 2 m/s, an error of 2 m/s in wind speed corre- sponds to an error of 100% on the emissions. It maximizes the local enhance- ments which helps with the large noise, but this filter is typically the opposite in most studies using satel- lite data (minimum wind speed). Removing noise will help removing that threshold which seems to be a simple but risky solution to reducing the noise in model-data residuals.*

**adjusted**

Please see our answer to the first mayor comment of the referee. In the new version of the manuscript we will not filter on the wind speed anymore. We found removing these filter criteria have no significant

effect on our results (see the attached Fig. 1).

**2  Technical Comments**

1. **P2  L1: and its transport in the atmosphere : Unclear.**

   **adjusted**

   We changed the sentence p2,l1 from:
   " These characteristics established CO as a tracer for air pollution and its transport in the atmosphere . . . "
   to
   " These characteristics established CO as a tracer for air pollution and transport processes in the atmosphere "

2. **P2  L20: Add references for the error sources**

   **adjusted** We added a reference to (Borsdorff et al., 2019) where we discussed the errors on the example of pollution transport from Tehran.

3. **P2  L30: Here the emission estimation changed by 0.42 Tg/yr 30 in only 2 years from 2014 to 2016, due to a change in the mobile emission model from mobile to moves. This means that the emission model changed and not the emissions. Has this change been confirmed or validated by other data?**

   **adjusted**

   We haven't found a validation or confirmation of this. However, it shows that the error bars on the emissions of the INEM inventory are big. To prevent confusions, we removed the sentence.

4. **P4  L5: The inventory is time dependent and accounts for the diurnal, week-to-week and monthly variations of the emissions How accurate are these cycles? Considering the overpass time is fixed, the mismatch can be explained by a difference in diurnal cycles in and out of the city. How was the inventory constructed? Does it include traffic counts? Are the other sectors using temperature-dependent relationship?**

   **adjusted**

   There are no sectors using temperature-dependent relationships in the inventory. We are adding the following sentence at p4, l5:
   " The weekly and daily mobile temporal profiles are derived from traffic counts in Mexico. The emissions inventory is described in Garcia et al. (2018). "
   We added the following sentence at p11,l10:
   " Another potential error source of our method are the accuracy of the week-to-week and monthly variations of the emissions in the INEM inventory considering the fixed overpass time of TROPOMI."

5. **P4  L29: Here, z is the mean elevation in the TROPOMI CO ground pixels and zref = 2240 m the reference altitude which is set to the elevation of Mexico City. This correction is unclear. The altitude used by TROPOMI is defined as a surface pressure. The altitude error depends on the difference between the WRF surface pressure and the TROPOMI surface pressure. Why using an average altitude of Mexico City as a reference?**

   **adjusted**

   Actually, the reference altitude can be arbitrary chosen. However, we selected a altitude that is in the range of our region of interest. We change the sentence at p4, l29 from:
   " Here, $z$ is the mean elevation in the TROPOMI CO ground pixels and $z_{\mathrm{ref}} = 2240$ m the reference altitude which is set to the elevation of Mexico City. "
   to
   " Here, $z$ is the respective elevation of the TROPOMI CO ground pixels and $z_{\mathrm{ref}} = 2240$ m is an arbitrary reference altitude set to the elevation of Mexico City, . . . "

6. **P5 - L4: local enhancements of CO are due to emissions of the city districts of the same day Have you tested that assumption? Basin cities are often problematic with low wind speed for days, which can accumulate signals from more than one day in the basins (example: Los Angeles during Winter). An averaged wind speed or residence time of tracers would help justify this assumption.**

   **adjusted**

   We added the following sentence at p11,l10: " Furthermore, basin cities can be problematic with low wind speed for days, which could lead to accumulate signals from more than one day in the basins which is not yet covered by our approach. To account for this effect in our inversion needs major adjustments, which will be investigated in follow up studies. "

7. **P5: Use the current notation for multivariate regression used in most publications (observation operator H, state vector x, prior error cov B, Obs error cov R, observations y, Kalman gain K).**

   **not adjusted**

   We are using the notation of Rodgers (2000) that is commonly used in our field.

8. **P6 - L17: with little forward model errors. Unclear. Re-phrase.**

   **adjusted**

   We changed the sentence p6,l17 from:
   " On one hand, it should be large enough to estimate mean emissions for the period of TROPOMI observations, and the other hand strict data filtering is required to get a stable inversion with little forward model errors"
   to
   " On the one hand, the ensemble should be large enough to estimate mean emissions for the considered time period, but on the other hand it should be strictly filtered for cases where the forward model is in good agreement with the measurement such that a stable inversion of the all emissions is possible. "

9. **P6 L24: for low wind speeds more reliable. The model errors are critical during low wind speed conditions when a slight change in wind speed can affect the magnitude of the observed enhancements. Typically, high wind speeds should be avoided because local enhancements are weak while low wind speeds should also be avoided when a small change in the wind speed can significantly change the local enhancements.**

   **adjusted**

   Please see our answer to the first mayor comment of the referee. In the new version of the manuscript we will not filter on the wind speed anymore. We found removing this filter criteria has no significant effect on our results (see the attached Fig. 1).

10. **P6 L30: fit residuum ¡ 8ppb, and the standard deviation ¡ 8ppb to limit the effect of too large forward model errors. TROPOMI is an averaged enhancement over a grid cell. Point sources will be under-estimated in the data as the plume will not be mixed over the entire grid cell. This bias has been presented by the TROPOMI team. Can you conform the relationship between point sources over the domain and the location of these high model-data differences?**
    **In addition, removing noisy pixels will artificially decrease the uncertainty by removing undesirable pixels. Some of these large model-data differences might be real transport errors or observation noise.**

    **adjusted**

    The under-estimation reported by the TROPOMI was analyzed for point-sources smaller than a TROPOMI pixel deploying the mass balance method. This does not hold here, we are looking at city districts that in general extend over multiple TROPOMI ground pixel (see Fig. 4 of the old manuscript).
    The filter criteria criticised by the referee is not applied anymore in the new version of the manuscript. Please have a look at our answer to the first major comment of the reviewer. Removing this criteria was not changing our results significantly (see the attached Fig. 1).

11. **P7 L1: ...(ymeas) ¡ 0.65 to ensure that the forward model can explain the vari- ability of the measured CO field. This value seems arbitrary. How did you define it?**

   **adjusted**

   Please see our answer to the first mayor comment of the referee. In the new version of the manuscript we will not apply this filter anymore. We found removing this filter criteria has no significant effect on our results (see the attached Fig. 1).

12. **P7 L4: What I the impact on the seasonal distribution? Does it remove data evenly over the year? This filters are likely to bias your results over specific season. A figure showing the time depencen of the filtered data is needed (or statistics)**

   **adjusted**

   Please see the major comment one of this referee. We removed many of this filters and our results are still stable. However, an even sampling over all seasons is not use full here. We described in p8l1-9 that in particular measurements during the biomass burning season are too difficult to interpret with our method and should be rejected to ensure data quality.

13. **P7 L15: Kbg and Kelv are not regularized How can you optimize the emissions without regularizing the background values? Are they pre-determined? How were they defined?**

   **adjusted**

   The signal about the background parameters is very strong. The parameters are fitted without regularization for each overpass of TROPOMI. Please see the major comment of referee 1.

14. **P7 15-20: The balance between prior information and data constraint is usually computed with the Chi2 normalized distance. A value near one will define the optimal balance between the two. Michalak, A., Hirsch, A., Bruhwiler, L., Gurney, K., Pe- ters, W., and Tans, P.: Maximum likelihood estimation of co- variance parameters for Bayesian atmospheric trace gas sur- face flux inversions, J. Geophys. Res.-Atmos., 110, D24107, https://doi.org/10.1029/2005JD005970, 2005.**

   **not adjusted**

   A major problem we are facing in this study is that the priori information from the INEM inventory seems to be biased. Hence, we don't want to constrain our inversion to much by it.

15. **P7 L21-24: This approach weighs toward pixels that are co-located with the sources. In other words, it selects preferentially the pixels above the city. In general, it should work but an evaluation perido would be helpful (with and without the filter) to measure the impact of the selection. This approach might bias the results if the model under- /over-estimate urban pixels.**

   **not adjusted**

   When we don't apply the regularisation the inversion depends on forward model errors what we showed in this study.

16. **P7 L23: temporal variation of the INEM emissions to be about 40%**

   . . .

   **vary with 60% around their average How do you define the 60%? The link between 4060% is not explained. In addition, temporal variations and mean emission errors are not supposed to scale together. This part needs to be described more carefully. The emission errors should depend on the emissions alone instead of their temporal vari- ability.**

   **adjusted**

   The value 60% was chosen to not over regularize the inversion. Hence, we could of course enforce 40% here but this choice is a balance between propagation of errors and extraction of information content from the measurement. To make this more clear, we change the paragraph at p7,l22 from:

" Considering the temporal variation of the INEM emissions to be about 40%, we adjusted the regularization parameter $\gamma_1, \cdots, \gamma_{10}$ such that the retrieved emissions vary with 60% around their average. Hence, our regularization is not enforcing that the retrievals show the same variability as the emissions of the inventory. The value 60% is an empirical parameter which the retrieval more freedom to balance information content and noise propagation. This puts a moderate constraints on the inversion ensuring on one hand a stable inversion and on the other hand a realistic variation of the retrieved emissions around the priori. "

to

" Considering the temporal variation of the INEM emissions to be about 40%, we adjusted the regularization parameter $\gamma_1, \cdots, \gamma_{10}$ such that the retrieved emissions vary within 60% around their average. The value 60% is empirical chosen to balance information content against noise propagation. It puts a moderate constraints on the inversion ensuring on the one hand a stable inversion and on the other hand a realistic variation of the retrieved emissions around the priori. "

17. *Figure 4: This figure provides illustrations but is not very helpful to prove that the WRF model is reliable or good enough. Instead, model-data mismatches should be pre- sented in a synthetic figure, for different times of year, using whisker boxes. Snapshots for four days out of 160 is too few to convince the readers. This figure should be replaced.*

**adjusted**

We disagree. We think that this examples clearly show the advantages and limitations of our our approach as we also discuss it in the manuscript.

18. *P8 L18: This clearly shows that regional models like WRF have a great potential for the interpretation and analysis of TROPOMI data. No, this does not demonstrate the model capabilities nor the ability fo the model to extract emissions. Re-phrase.*

**adjusted**

We changed the sentence p8,l18 from:
"This clearly shows that regional models like WRF-chem have a great potential for the interpretation and analysis of TROPOMI data. "
to
" This clearly shows that regional models like WRF-chem have a great potential to reproduce the large-scale patterns seen by the TROPOMI instrument. "

19. *P8 L21: For atmospheric conditions under high wind speeds the WRF simulations can deviate more from the TROPOMI measurements as shown in Fig. 4 (c). This single day is too limited to conclude anything. More statistics on windy days are needed to prove your point is valid here.*

**adjusted**

The aim of the study is not to analyze model error under high wind speeds. Please have a look at our answer of the major comment of referee 1. We removed the filtering on wind speed and our results were not changing significantly (see the attached Fig.1)

20. *Figure 5: The modeled and observed XCO should be presented first, summarized for the days available before and after filtering. Do the residuals show a seasonality? The results of the optimization are difficult to interpret without the evaluation of the initial model results.*

**not adjusted**

The comment is not well formulated and we don't understand it.

21. *P9 L1-6: Large model-data mismatches are expected from observations, model er- rors, and prior errors. If observations are being used twice (if I understood correctly), mismatches will decrease automatically. Optimization should never be performed a second time with the same data. Unless I misunderstood the approach (different data are being used between these two steps), only one step should be performed. Other- wise the constraint from the data is over-estimated.*

**adjusted**

Please see our answer to the major comment two of the referee.

22. ***P9 L7: Furthermore, non-uniform variation of the background CO concentration can be a additional reason for this scatter (as shown in Fig. 3). How did you determine the background? How do you separate the contribution from the city emissions? Is it all performed within the inversion? If so, how do you define background uncertainties? Some additional tests should be performed. If you introduce a background in the bias, is your optimization system able to recover that bias? Figure 6: statistics should be presented for the entire data set and not only for four days.***

**adjusted**

The background parameters are retrieved together with the emission estimates from the measurements of each TROPOMI overpass. Please have a look at the major comment of referee 1 were we adapted the manuscript accordingly. Fig. 2 of the old manuscript shows for each TROPOMI measurement the inverted background values together with the retrieval errors represented as error bars. However, it is for sure also important to analyze individual cases like shown in Fig.5 because not every issue becomes visible when looking at statistics. We tested our inversion routines and they are working. Hence, introducing and retrieving an artificial background is not helping further here.

We added a sub-figure in Fig.6 showing the goodness of the fit between TROPOMI and WRF for the priori emissions, the ones of the Pre-fit and Final. We changed the paragraph p6, l3-8 from:

" This yields the mean

$$\langle \delta \rangle = \frac{1}{J} \sum_{j=1}^{J} \delta_j \tag{13}$$

and the standard deviations of the residuals

$$\sigma(\delta) = \frac{1}{J-1} \sum_{j=1}^{J} (\delta_i - \langle \delta \rangle)^2 \tag{14}$$

The standard deviation $\sigma(\mathbf{y}_{\mathrm{meas}})$ and $\sigma(\mathbf{F}(\mathbf{x}))$ of the TROPOMI CO field and the corresponding WRF-chem forward simulation completes our set of diagnostics.
"
to

" To evaluate the fit quality for each overpass, we consider the fit residuals $\delta_i = y_i - \mathbf{K}_i x_{\mathrm{est},i}$. Additionally, we evaluate the goodness of the fit described by the reduced chi squared value,

$$\chi_i^2 = \frac{1}{\nu_i} \sum_{l=1}^{L} (\delta_i, l/y_{\mathrm{err},ik})^2 \ . \tag{22}$$

Here $L$ is the number of observation of a single overpass, $y_{\mathrm{err},i}$ the retrieval error, and $\nu_i = L - \mathrm{DFS}_i$. "
We changed the caption of Fig.6 from:
" ... as a robust estimation of the standard deviation and the number of collocations (c). The number of collocation of the Pre-fit is the same for all tracer domains (blue line) but in the Final-fit it is changing due to the information content filtering. Here, a collocation corresponds to a specific day because TROPOMI overpasses the region only once.
"
to
" ... as a robust estimation of the standard deviation, (c) the median of the goodness of the fit ($\chi^2$), and the number of collocations (d). The number of collocations and the $\chi^2$ values of the apriori simulation and Pre-fit are the same for all tracer domains (blue and grey line) but in the Final-fit it is changing due to the information content filtering. Here, a collocation corresponds to a specific day because TROPOMI overpasses the region only once. "
We added the discussion at p9,l9:
" The $\chi^2$ values in Fig. 6 clearly show that the agreement between TROPOMI and WRF can be improved by fitting the emissions of the different city districts (blue line) instead of using the INEM inventory (grey

line). The regularization approach increases the $\chi^2$ values (green bars) because the inversion can less compensate differences between TROPOMI and WRF by choosing unrealistic emissions. However, the $\chi^2$ values of the Final-Fit are still lower than the ones for the prior INEM emissions (grey line). Overall, the $\chi^2$ values exceeds 1 which indicates that the difference between TROPOMI and WRF is dominated by systematic errors in the WRF simulation. "

23. ***P9 L13-14: the averaging kernel shows that the Final-fit inversion is insensitive to deviations of the Tulancingo emission from the prior estimate. Whereas the Pre-fit inversion estimates very small emissions for this district, the subsequent regularization changes the emission only marginally. This is the direct consequence of performing an optimization with the same data twice. Some of the constraint has already been introduced in the emissions.***

    **adjusted**

    Please also see our answer to the major comments of the referee. The idea of the Pre-fit is to estimate mean emissions for all regions to prevent biases due to imposing a regularization. Hence, we want that the emissions of the Pre-fit and Final are the same.

    We change the sentence at p9, l14 from:
    " Moreover, the averaging kernel shows that the Final-fit inversion is insensitive to deviations of the Tulancingo emission from the prior estimate. The reason for this is that the Pre-fit inversion was only estimating very small emissions for this district and the regularization of the Final-fit is changing this only marginally. "
    to
    " It indicates that TROPOMI measurements can be used to distinguish emissions of the different urban districts of Mexico, with the exception of the emissions of district Tulancingo. Due to the small mean emission, the averaging kernel indicates a low sensitive of the data product. "

**3 Special Comments**

1. ***Specific comments: P6 L15: depends crucially = highly depends on***

    **corrected**

**References**

Borsdorff, T., aan de Brugh, J., Pandey, S., Hasekamp, O., Aben, I., Houweling, S., and Landgraf, J.: Carbon monoxide air pollution on sub-city scales and along arterial roads detected by the Tropospheric Monitoring Instrument, Atmospheric Chemistry and Physics, 19, 3579–3588, https://doi.org/10.5194/acp-19-3579-2019, URL https://www.atmos-chem-phys.net/19/3579/2019/, 2019.

Rodgers, C. D.: Inverse methods for atmospheric sounding: theory and practice, vol. 2 of *Series on atmospheric, oceanic and planetary physics*, World Scientific, Singapore, River Edge, N.J., reprinted : 2004, 2008, 2000.

[Figure]

Figure 1: Statistics of CO emissions averaged from the 9th of November 2017 to the 25th of August 2019 for the tracer domains shown in Fig.1. (a) Median of the priori emissions (adapted INEM inventory) used for the WRF-chem simulation (grey) and retrieved from the TROPOMI data (Pre-fit in blue, Prefit-fit with relaxed filtering in green). The error bars indicate the standard error of the mean calculated from the delta percentiles (b) used as a robust estimation of the standard deviation and the number of collocations (c). The number of collocation of the Pre-fit is the same for all tracer domains (blue and green line). Here, a collocation corresponds to a specific day because TROPOMI overpasses the region only once.

---

## Author Response (AR2)

**author comments on the manuscript "Monitoring CO emissions of the metropolis Mexico City using TROPOMI CO observations", editor**

We would like to thank the editor. In this document we provide our replies to the editor's comments. The original comments made by the editor are numbered and typeset in italic and bold face font. Following every comment, we give our reply. Here line numbers, page numbers and figure numbers refer to the original version of the manuscript, if not stated differently. Additionally, the revised version of the manuscript is added.

**1 Minor Comments**

1. ***Reviewer1, point 6: I agree that a comparison to FTIR is out of scope but you might want to give a statement on the agreement between FTIR and TROPOMI found in previous studies***
   **adjusted**
   We added the following statement with a reference to our recent TROPOMI CO validation deploying CO retrievals from ground-based FTIR measurements (p10,l8):
   " In general, the TROPOMI CO data product agrees very well with retrievals from ground-based FTIR measurements performed by the TCCON network world wide with an averaged bias less than $6 \pm 3.8$ppb and the bias with retrievals from NDACC measurements is even lower (Borsdorff et al., 2018). "

2. ***Reviewer 2, point 20: I interpret the comment by the reviewer as a request for a time series showing modelled and observed CO to diagnose the model-observation mismatch. I would like to suggest to consider including such a figure.***
   **adjusted**
   We added a new figure showing the temporal variability of the mismatch between TROPOMI CO and the WRF-chem simulations for all retrieval cases. In addition we added an discussion of the figure at (p10,l31):

[revised manuscript text omitted]